# Senotherapeutic drugs for human intervertebral disc degeneration and low back pain

Hosni Cherif[1,2], Daniel G Bisson[1,2]*, Matthew Mannarino[1,2], Oded Rabau[2,3]*, Jean A Ouellet[2,3], Lisbet Haglund[1,2,3]*

[1]Orthopaedic Research Lab, Department of Surgery, McGill University and the Research Institute of the McGill University Health Centre, Montreal, Canada; [2]McGill Scoliosis and Spine Group, Department of Surgery, McGill University and the Research Institute of the McGill University Health Centre, Montreal, Canada; [3]Shriner's Hospital for Children, 1003 Decarie Blvd, Montreal, Canada

**Abstract** Cellular senescence is a contributor to intervertebral disc (IVD) degeneration and low back pain. Here, we found that RG-7112, a potent mouse double-minute two protein inhibitor, selectively kills senescent IVD cells through apoptosis. Gene expression pathway analysis was used to compare the functional networks of genes affected by RG-7112, a pure synthetic senolytic with o-Vanillin a natural and anti-inflammatory senolytic. Both affected a functional gene network related to cell death and survival. O-Vanillin also affected networks related to cell cycle progression as well as connective tissue development and function. Both senolytics effectively decreased the senescence-associated secretory phenotype (SASP) of IVD cells. Furthermore, bioavailability and efficacy were verified ex vivo in the physiological environment of degenerating intact human discs where a single dose improved disc matrix homeostasis. Matrix improvement correlated with a reduction in senescent cells and SASP, supporting a translational potential of targeting senescent cells as a therapeutic intervention.

**\*For correspondence:**
daniel.g.bisson@gmail.com (DGB);
matthew.mannarino@mail.mcgill.
ca (MM);
odedrabau@gmail.com (OR);
lisbet.haglund@mcgill.ca (LH)

**Competing interests:** The authors declare that no competing interests exist.

## Introduction

Low back pain is a global health problem that is experienced by ~80% of individuals at some point in their lifetime (*Vos et al., 2012*). This problem is the number one single cause of years lived with disability with enormous personal and health system related costs (*Institute of Medicine (US) Committee on Advancing Pain Research, Care, and Education, 2016*; *Hartvigsen et al., 2018*). Intervertebral disc (IVD) degeneration is a major factor contributing to low back pain (*Vos et al., 2012*; *Adams and Hutton, 1983*; *Vergroesen et al., 2015*). The cellular pathogenesis of IVD degeneration and the mechanisms leading to pain are not fully understood. One novel approach to treat painful degeneration is to target cellular senescence, a state of irreversible growth arrest occurring in response to cellular stress (*Tchkonia et al., 2013*). Stress-induced premature senescence is caused by factors such as oxidative and genotoxic stresses (*Toussaint et al., 2000*; *Campisi and d'Adda di Fagagna, 2007*). Increasing evidence suggests that accumulation of senescent cells during tissue degeneration contributes directly to initiation and development of musculoskeletal degenerative diseases like osteoarthritis (*Jeon et al., 2017*) and IVD degeneration (*Le Maitre et al., 2007*; *Feng et al., 2016*; *Wang et al., 2016*; *Patil et al., 2018*; *Cherif et al., 2019*). Senescent cells secrete a range of cytokines, chemokines, growth factors, and proteases termed as the senescence-associated secretory phenotype (SASP) (*Herbig et al., 2006*; *Kuilman et al., 2008*; *Coppé et al., 2008*; *Xu et al., 2015*). These SASP factors are suggested to further induce senescence in a paracrine manner (*Acosta et al., 2013*), to promote matrix catabolism and sterile inflammation in IVDs, thereby

**eLife digest** Pain in the lower back affects about four in five people during their lifetime. Over time, the discs that provide cushioning between the vertebrae of the spine can degenerate, which can be one of the major causes of lower back pain.

It has been shown that when the cells of these discs are exposed to different stress factors, they stop growing and become irreversibly dormant. Such 'senescent' cells release a range of proteins and small molecules that lead to painful inflammation and further degeneration of the discs. Moreover, it is thought that a high number of senescent cells may be linked to other degenerative diseases such as arthritis.

Current treatments can only reduce the severity of the symptoms, but they cannot prevent the degeneration from progressing. Now, Cherif et al. set out to test the effects of two different compounds on human disc cells grown in the laboratory. One of the molecules studied, RG-7112, is a synthetic drug that has been approved for safety by the US Food and Drug Administration and has been shown to remove senescent cells. The other, o-Vanillin, is a natural compound that has anti-inflammatory and anti-senescence properties.

The results showed that both compounds were able to trigger changes to that helped new, healthy cells to grow and at the same time kill senescent cells. They also reduced the production of molecules linked to inflammation and pain.

Further analyses revealed that the compounds were able to strengthen the fibrous matrix that surrounds and supports the discs. Cherif et al. hope that this could form the basis for a new family of drugs for back pain to slow the degeneration of the discs and reduce pain. This may also have benefits for other similar degenerative diseases caused by cell senescence, such as arthritis.

accelerating the degenerative process (*Parrinello et al., 2005*; *Tominaga, 2015*). Elimination of senescent cells enhances disc tissue homeostasis in genetically modified progeroid Ercc1$^{-/\Delta 22}$ and p16-3MR (*Patil et al., 2019*) mice suggesting that senotherapeutic drugs have great potential to treat low back pain resulting from IVD degeneration. The effect could potentially be mediated by apoptotic (senoptosis) or nonapoptotic (senolysis) mechanisms (*Schmitt, 2017*; *Soto-Gamez and Demaria, 2017*; *Kirkland et al., 2017*; *Niedernhofer and Robbins, 2018*) or by modulating the SASP, indirectly suppressing senescence (senomorphics) (*Zhu et al., 2015*; *Soto-Gamez and Demaria, 2017*; *Kirkland et al., 2017*; *Childs et al., 2017*). Further, interest is growing toward the use of natural senotherapeutic compounds such as quercetin, fisetin and piperlongumine, curcumin and o-Vanillin (*Cherif et al., 2019*; *Li et al., 2019*); their key advantages being low toxicity and great potential to be translated into clinical applications.

Cell-cycle arrest of senescent disc cells is mainly mediated by the two pathways: p53-p21-Rb and p16-Rb. During disc degeneration, both pathways are activated simultaneously to induce senescence (*Feng et al., 2016*). The FDA-approved drug RG-7112 (RO5045337) (*Weber, 2010*; *Laberge et al., 2018*) is a highly potent and selective MDM2 antagonist (*Weber, 2010*; *Tovar et al., 2013*) that restores the physiological activity of p53. RG-7112 is the first nutlin family member to be assessed clinically (*Weber, 2010*; *Laberge et al., 2018*) showing evidence of acceptable safety (*Ray-Coquard et al., 2012*; *Constantinidou et al., 2012*). RG-7112 was reported to selectively kill senescent lung fibroblasts (IMR90) where senescence was induced by ionizing radiation (IR) (*Weber, 2010*; *Laberge et al., 2018*). The natural compound o-Vanillin, known for its antioxidant and anti-inflammatory effects (*Santosh Kumar et al., 2002*; *Oliveira et al., 2014*; *Shah et al., 2019*; *Marton et al., 2016*), has recently been described for its senotherapeutic activity in human IVD cells (*Cherif et al., 2019*).

One caveat in the field is that there are currently no large bipedal animal models that mimic naturally occurring IVD degeneration and human genetic background (*Alini et al., 2008*; *Jin et al., 2018*). One solution to this caveat is the use of an intact disc culture model (*Gawri et al., 2011*; *Krock et al., 2014*; *Gantenbein et al., 2015*; *Rosenzweig et al., 2016*) of human IVDs to test and develop new senotherapeutic drugs suitable for human disc degeneration.

Here, we utilized in vitro and ex vivo models, to assess the senotherapeutic effects of RG-7112 and o-Vanillin on naturally occurring senescent cells in degenerating human IVDs. We have

previously shown that o-Vanillin has senolytic effects on senescent human IVD cells. o-Vanillin is a natural compound that in addition to its senolytic effect has antioxidant and anti-inflammatory properties. We compared the effect of o-Vanillin to RG-7112, a pure senolytic drug, that we found to efficiently kill senescent annulus fibrosus (AF) and nucleus pulposus (NP) cells. The objective was to evaluate if o-Vanillin with its dual function could further reduces inflammatory factors released by non-senescent cells thus enhancing the therapeutic effect. We also aimed to establish if one or both drugs could reach and kill senescent IVD cells in their native environment of intact human disc with naturally occurring degeneration.

## Results

### Determination of the senotherapeutic activity of RG-7112

We aimed to compare the senotherapeutic activity of the natural senolytic o-Vanillin a natural senolytics with antioxidant and antiinflammatory properties with a pure synthetic senolytic drug. We chose the commercially available FDA approved drug RG-7112 as a candidate. A concentration of 5 µM RG-7112, shown as a safe and effective dose for human IMR90 lung fibroblasts, was used to evaluate its effect on senescent and non-senescent human IVD cells (*Weber, 2010*; *Laberge et al., 2018*). Pellet cultures of IVD cells from degenerate NP and AF regions were exposed to RG-7112 for 4 days, the drug was removed, and the treated pellets were maintained for 21 days in standard media. Then, evaluated for cytotoxicity, senolytic and therapeutic activity. No cytotoxicity was observed following treatment. In contrast, both NP (9.87%; p<0.05) and AF (11.87%; p<0.05) cells showed a significant increase in metabolic activity (*Figure 1—figure supplement 1A*). Pellet cultures were treated as before and the presence of p16$^{Ink4a}$, Ki-67 and caspase-3 were analyzed after 21 days (*Figure 1A*). RG-7112 significantly decreased the percentage of p16$^{Ink4a}$ positive NP (11.48%; p<0.001) and AF (20.03%; p<0.0001) cells compared to untreated control cultures (*Figure 1B*). Similarly, RG-7112 significantly increased the percentage of Ki-67-positive NP (23.07%; p<0.05) and AF (34.92%; p<0.0001) cells (*Figure 1C*). Caspase-3-positive AF cells increased significantly (17.88%; p<0.01) while a non-significant increase (2.94%; p=0.51) was observed in NP cells following treatment (*Figure 1D*). Caspase 3 and caspase 3/7 activity was confirmed by fluorescence microscopy and activity assays, respectively (*Figure 1 —figure supplement 1B-C and d-f*). Confocal immunofluorescence confirmed co-localization of p16$^{Ink4a}$ and caspase-3, whereas proliferating (Ki-67-positive) cells did not co-localize with p16$^{Ink4a}$ (*Figure 1 —figure supplement 1E* (a–c). RG-7112 selectively increased apoptosis (*Figure 1 —figure supplement 1E*) while it maintained comparable metabolic activity (*Figure 1 —figure supplement 1F*) in NP cells from both degenerate and non-degenerate IVDs. Finally, we investigated proteoglycan synthesis using the DMMB assay following treatment. A significant increase in proteoglycan release in conditioned media was observed after 14 days of treatment (*Figure 1 —figure supplement 1G*).

### Gene expression analysis revealed the potential pathways by which RG-7112 and o-Vanillin mediate their senolytic effect

To identify molecular pathways affected by RG-7112 and o-Vanillin in human disc cells, gene expression analysis of 96 selected cellular senescence genes was performed and 91 of the 96 genes were expressed at a detectable level. Pellet cultures were treated with RG-7112 (5 µM) or o-Vanillin (100 µM) for 4 days. The relationship between the differentially up- and downregulated genes are depicted in *Figure 2A*. Compared with the control group, the number of differentially expressed genes for o-Vanillin were 40 (30 upregulated and 10 downregulated) and eight for RG-7112 (6 upregulated and two downregulated). In upregulated DEGs, three are common to both drugs, mitogen-activated protein kinase 14 (MAPK14), cell division cycle 25 c (CDC25c) and cyclin dependant kinase 2D (CDKN2D or p19$^{ARF}$). No down-regulated DEGs were common to the drugs. Although cyclin B1 (CCNB1) is significantly expressed following the treatment with the two compounds, it is upregulated in RG-7112 and downregulated in o-Vanillin (*Figure 2A*). Of the 91 genes identified, 44 were differentially expressed with a p<0.05 in one or both treatments (*Figure 2B*). The 47 genes that did not meet the significance criteria of p<0.05 are shown in (*Figure 2—figure supplement 1A*). Next, we compared gene expression profiles of the o-Vanillin group with the RG-7112 group. Of the 91 evaluated, only eight genes were significantly affected by RG-7112 treatment, four were common to

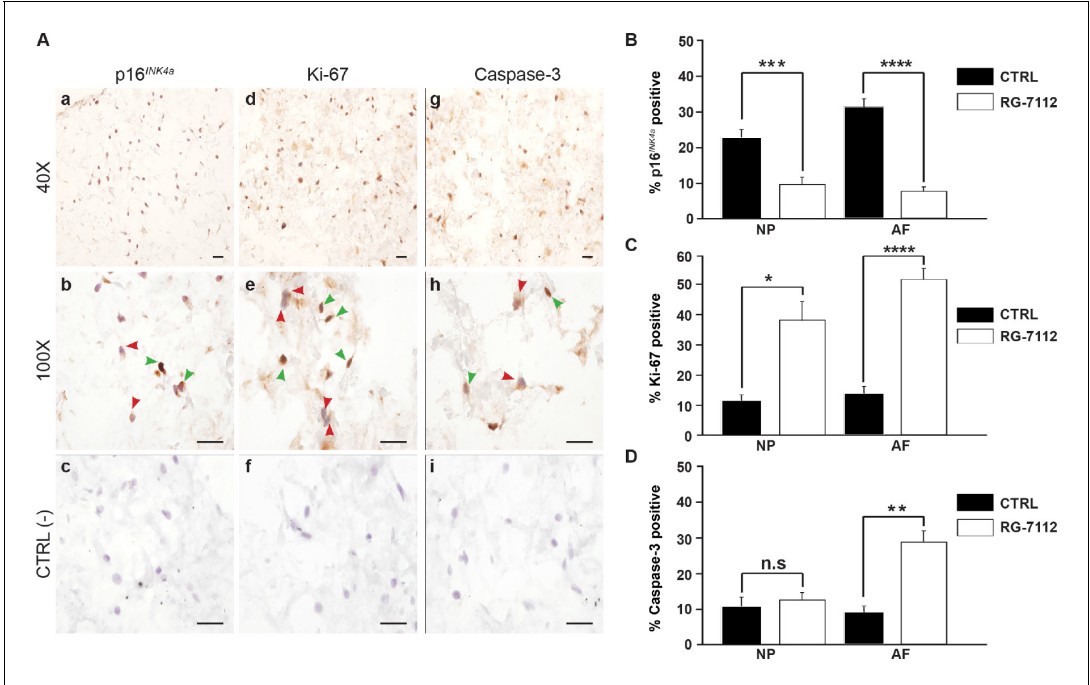

**Figure 1.** RG-7112 treatment of IVD pellet cultures. (**A**) Representative images indicating how we deemed positive and negative p16$^{INK4a}$ (a), Ki-67 (d), and caspase-3 (g) staining. (b, e and h) Magnified images of (a, d and g). Arrow heads indicate positive (green) and negative (red) staining. (c, f and i) no secondary antibody control photomicrographs. Quantification of (**B**) p16$^{INK4a}$ (n = 7), (**C**) Ki-67, and (**D**) caspase-3 expression, (n = 8). Scale bars: 20 µm in (**A**). The cells were from degenerating IVDs as indicated in *Table 2*. Values are presented as mean ± SEM in (**B–D**). * Indicates a significant difference assessed by the two-tailed Student's t-test: p<0.05; **p<0.01 and ****p<0.0001.

The online version of this article includes the following source data and figure supplement(s) for figure 1:

**Source data 1.** Excel file containing the percentage of p16$^{Ink4a}$, ki-67 and caspase three positive in NP and AF cells control and treated with RG-7112.
**Figure supplement 1.** Senolytic activity of RG-7112 on human IVD cells in vitro.
**Figure supplement 1—source data 1.** Excel file containing the fluorescence values for metabolic and caspase3/7 activity, the number of apoptotic cells, and the concentrations of sGAG in NP (and AF) cells treated and untreated with RG-7112.

o-Vanillin (MAPK14, CCNB1, CDC25c and CDKN2D (p19$^{ARF}$)), and four were exclusive to RG-7112 (MDM2, CDKN1A (p21$^{ARF}$), E2F1 and RBL1). In contrast, 40 genes were significantly affected by o-Vanillin treatment. Cell cycle and senescence genes were significantly downregulated including cyclin dependent kinase 2A (CDK2A or p16$^{Ink4a}$), cyclin dependent kinase 2C (p18$^{ARF}$), Cyclin A2, CCNB1, CDC25c, Vimentin, Mitogen-activated protein kinase 6 (MAPK6) and Checkpoint kinase 1 (CHEK1) (*Figure 2C*). Examples of apoptotic and proliferative genes significantly upregulated by o-Vanillin include B-cell lymphoma 2 (Bcl-2), Bcl-2-like 1, Bcl-2-like 2, interferon regulatory factor 5 (IRF5), IRF7 and receptor tyrosine-protein kinase ERBB2. Interestingly, MAPK14, CDC25c and CDKN2D (p19$^{ARF}$) were the only genes significantly upregulated after the treatment with both compounds, while CCNB1 showed opposite regulation patterns. In conjunction, apoptotic pathways are significantly upregulated to kill senescent cells while proliferation related pathways are activated in non-senescent cells. Detailed fold change difference of differentially expressed genes and respective p value are included in *Supplementary file 3 (a-d)*.

To gain a further insight into a potential mechanism of action, the differentially expressed genes were mapped to networks in the Ingenuity Pathways Analysis (IPA) database. The scores take into account the number of focus genes and the size of the network to approximate the relevance of the network to the original list of focus genes. The IPA core analysis features allowed identification and determination of one network connecting the cell cycle, cell death and survival, connective tissue development and function pathways (RG-7112, network 1) of NP cells treated with RG-7112 (*Figure 2D*). In the o-Vanillin-treated pellets, the highest-scoring network revealed a significant link with cell death and survival, neurological disease and organismal injury and abnormalities (o-Vanillin, network 1) (*Figure 2E*). Furthermore, connective tissue development and function, cell cycle (o-

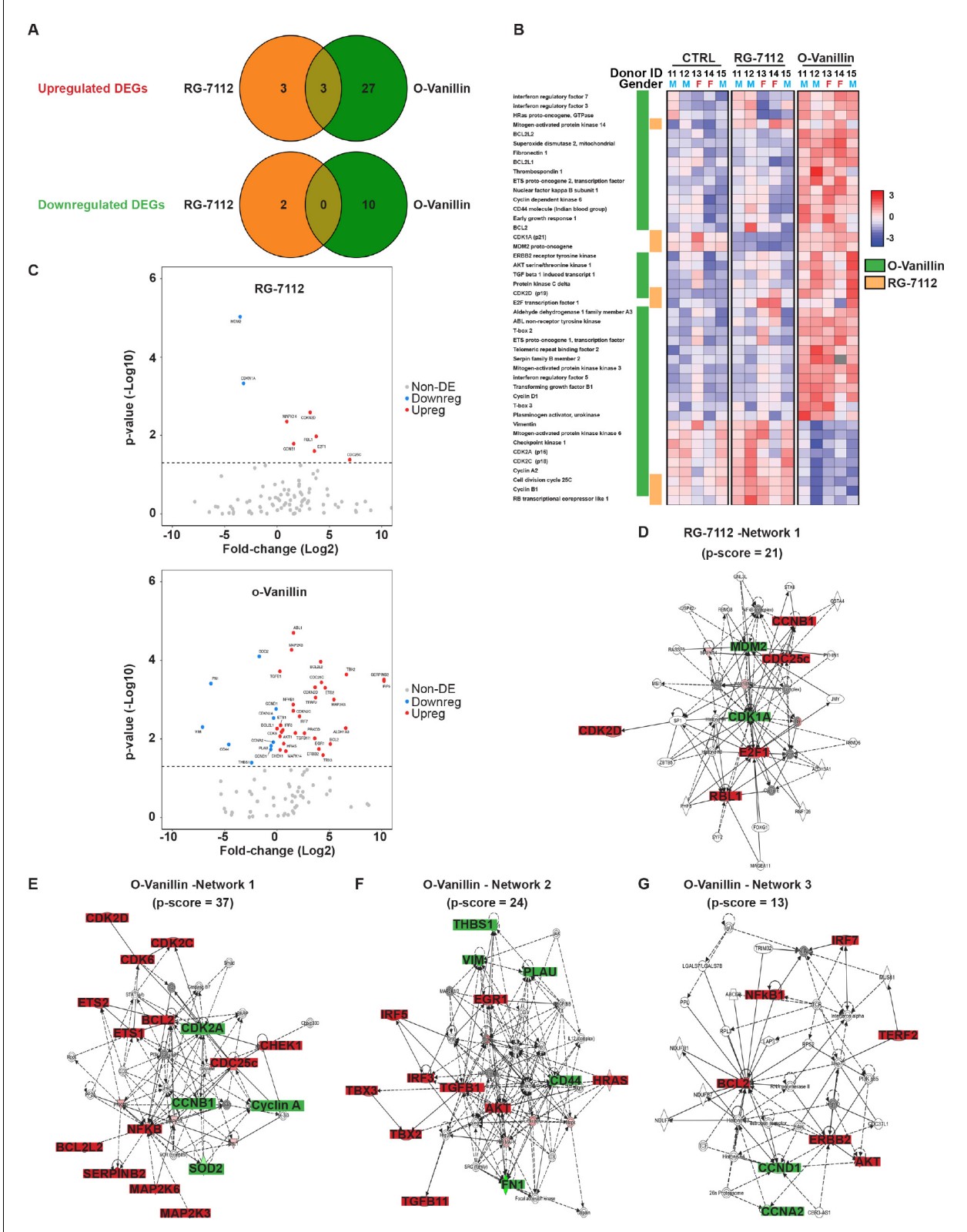

**Figure 2.** Differentially expressed senescence related genes in NP pellets. (**A**) Venn diagrams of the differentially up and downregulated genes among the different groups. O-Vanillin, RG-7112-treated NP cells in pellets culture. For Upregulated genes odds ratio (OR) = 2.13 and p=0.39; for Downregulated genes: OR = 0 and p=1. (**B**) Heatmap of the top 44 over and under expressed genes in control (CTRL), RG-7112 and o-Vanillin-treated NP cells. All genes shown were first normalized to the housekeeping gene GAPDH. Data shown are relative to the calculated Z scores across the

*Figure 2 continued on next page*

*Figure 2 continued*

samples (see Materials and methods) and ranked by significance adjusted to p<0.05. Red represents relatively high levels of expression; blue represents relatively low levels of expression. Significantly differentially expressed genes are indicated with green lines for o-Vanillin and in orange lines for RG-7112. Each column represents one individual (for a total of n = 5 per group) and each row represents expression of a single gene. Donor ID and gender are indicated for each subject. (C) Volcano plots of mRNA expression of o-Vanillin and RG-7112 treated NP pellets: Plotted along the x-axis is the mean of log2 fold-change, along the y-axis the negative log10 of the p-values. Blue circles refer to downregulated genes, red circles refer to upregulated genes and grey circles to non-DEGs in o-Vanillin and RG-7112-treated NP pellets. The horizontal grey line is the negative logarithm of the t-test-adjusted p-value threshold (-log10 of p=0.05). (D) IPA diagrams of differentially expressed genes in RG-7112 and (E–G) o-Vanillin-treated NP pellets within the selected set of 91 genes. Direct and indirect interactions are shown by solid lines and dashed lines respectively. Green indicates gene downregulation; red depicts upregulation and molecules found by the data mining tools of IPA (build tools) are shown in gray. Color intensity represents the average of log2 fold change with brighter colors representing a more significant difference between treated and controls. Symbols for each molecule are presented according to molecular functions and type of interactions. Functional assignations attributed by IPA software. Significant difference set at p<0.05 was assessed by repeated measures Analysis of Variance (ANOVA) with Turkey's post hoc test for multiple pairwise comparison in (B–C) and Fisher's exact test in (A, D–G). The cells were from degenerating IVDs as indicated in *Table 2*.

The online version of this article includes the following source data and figure supplement(s) for figure 2:

**Source data 1.** Excel file containing the list of Differentially Expressed genes significantly up- or downregulated at p < 0.05 in RG-7112 and o-Vanillin conditions and the raw values of the fold change in control and treated groups for all tested genes.

**Figure supplement 1.** Non differentially expressed genes Heatmap and NP area measure in human IVD.

**Figure supplement 1—source data 1.** Excel file containing the the list of Non-Differentially Expressed genes in RG-7112 and o-Vanillin conditions and the raw values of the fold change of the 91 genes tested in control and treated groups.

Vanillin, network 2), cancer, cellular movement (o-Vanillin, network 3), were shown to be influenced in the other two networks (*Figure 2F–G*). All networks were identified and ranked by the score of the calculated p-value of the IPA assay within the selected set of 91 genes, the scores and molecules used to order these networks are shown in *Supplementary file 4 (a-b)*. These results confirmed the expected mode of action for RG-7112 and provide new insights to the predicted pathways involved in senescence and the responses to treatment with o-Vanillin.

## RG-7112 and o-Vanillin reduced inflammatory factors

Transcriptomic results following the treatment with the two senolytics predicted the activation of apoptotic (in senescent cells) and proliferative (in non-senescent cells) pathways which suggest a reduction in the senescence burden likeley affecting SASP factor release. To verify an effect on SASP, media of untreated NP cell pellets, RG-7112 and o-Vanillin-treated pellets were used. The media collected were from the same donors used for gene expression analysis in *Figure 2*. Samples were analyzed using an antibody array that simultaneously screen 80 factors. Out of the 80 factors present, 50 were detected in at least four donors and were included in further analysis (*Figure 3 — figure supplement 1A*). The 25 most affected SASP factors are divided into four classes: Cytokines (*Figure 3A–a*), chemokines CC (*Figure 3A–b*), chemokines CXC (*Figure 3A–c*), growth and neurotrophic factors (*Figure 3A–d*). The most reduced factors following RG-7112 treatment compared with control were for cytokines (TNF-α; −78.05 ± 4.69%) (p=0.71), CC-chemokines (CCL26; −66.69 ± 7.09%) (p=0.93), CXC-chemokines (CXCL13; −29.02 ± 17.26%) (p=0.68), growth and neurotrophic factors (HGF; −65.24 ± 5.9%) (p=0.87). The top four factors decreased by o-Vanillin treatment were: IL-7 (−60,37 ± 11.46%) (p=0.5), CCL26 (−40,14 ± 3.41%) (p=0.99), CXCL10 (−18,11 ± 6,15%) (p=0.62) and HGF (−52,42 ± 5,45%) (p=0.96) (*Figure 3A* (a-d)). To better visualize the overall effect of the two drugs on SASPs release, a scatter plot of average changes was generated demonstrating a similar overall effect of the two senolytics (*Figure 3A–e*). The trends for all 80 factors are visualized in (*Figure 3—figure supplement 1B-C*). The semi-quantitative cytokine array results were used to select 19 factor that were quantified with the Luminex assay. The 19 factors were selected based on the following criteria: only factors detected in all conditions and in at least four donors were chosen. These factors were selected to represent all the SASP classes and to include the most variably expressed cytokines, chemokines and growth factors. The 19 factors selected for further investigation were (INF-γ, TNF-α, IL1-α, IL1-b, IL-6, IL-8, CCL5, CCL7, CCL11, CCL24, CCL26, CXCL1, CXCL5, CXCL9, CXCL10, CXCL11, CX3CL1, Angiogenin and VEGF-A). The concentrations were measured and expression heatmap showed an overall decrease in the level of proinflammatory factors from both RG-7112 and o-Vanillin when compared to non-treated cultures. (*Figure 3B*). The levels of INF-

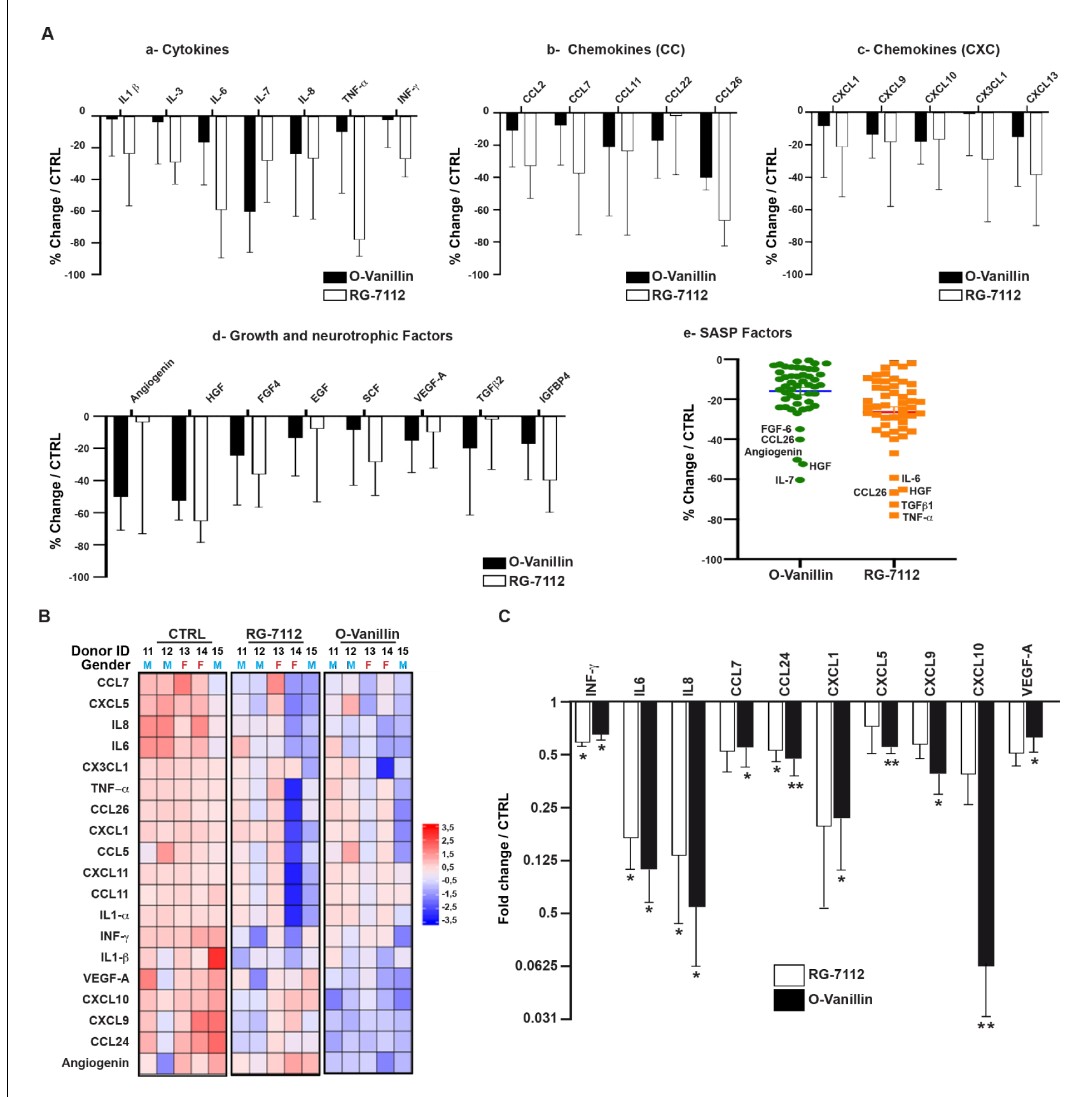

**Figure 3.** Profile of SASP factors released from pellet cultures following senolytic treatment. (**A**) Culture media was analyzed by a RayBio Human Cytokine Array. Relative mean densitometry units of the 80 factors were normalized to untreated controls with the most 25 downregulated SASP factors presented: cytokines (**A–a**), CC-chemokines (**A–b**), CXC-chemokines (**A–c**), growth and neurotrophic factors (**A–d**). Scatter plot showing the distribution in average change of 50 cytokines quantified using cytokine array (**A–e**). (**B**) Heatmap displaying quantification of 19 selected cytokines (19-plex Luminex array). Each column represents one individual (n = 5). The rows represent expression of a single protein. Data shown are log2 (fold change) relative to the average expression level in each condition. Donors ID and gender are indicated for each subject. (**C**) Significantly downregulated factors are presented as mean fold difference ± SEM; (n = 5). Culture media was collected from the same NP cells used in *Figure 2*. *Indicates significant difference assessed by repeated measures Analysis of Variance (ANOVA) with Turkey's post hoc test for multiple pairwise comparison: p<0.05 and **indicates p<0.01.

The online version of this article includes the following source data and figure supplement(s) for figure 3:

**Source data 1.** Excel file containing the meausres for the selected and significantly SASPs factors measured using cytokine array and Luminex analysis in NP pellets media following the treatment with RG-7112 and o-Vanilin.

**Figure supplement 1.** Total cytokine array and Luminex measures in pellet culture media.

**Figure supplement 1—source data 1.** Excel file containing the complete list of SASP factors measured by cytokines arrays and the non-significantly regulated factors measured by Luminex assay in treated NP pellets media.

γ, IL-6, IL-8 and CCL24 were significantly decreased following both treatments whereas the decrease in CCL7, CXCL1, CXCL5, CXCL9, CXCL10 and VEGF significance was reached only following o-Vanillin treatment (*Figure 3C*). However, significance was not reached for TNF-α, IL1-α, IL1-β, CCL5, CCL11, CCL26, CXCL11, CX3CL1 and Angiogenin in treated cultures compared to untreated

cultures (*Figure 3—figure supplement 1C*). Although, o-Vanillin reduced more SASP factors than RG-7112 at a significant level when compared to untreated cultures, there was no significant difference in concentrations between o-Vanillin and RG-7112-treated cultures, validating the similar overall effect the two drugs have on SASP factor release observed by cytokine array in *Figure 3A–f*. These results demonstrate that both RG-7112 and o-Vanillin decreased the overall inflammatory environment in pellet cultures of degenerating tissue and suggest a broader anti-inflammatory effect of o-Vanillin treatment.

## Senotherapeutic treatment improves disc matrix and reduces senescent cells in intact human disc

RG-7112 or o-Vanillin was injected into the central region of intact human IVDs to verify that the drugs can reach and kill the target cells in native tissue. Discs were pre-cultured for 4–6 days. The IVDs were treated with a single injection of vehicle, o-Vanillin, or RG-7112, delivered to the centre of the NP and were cultured for another 28 days as outlined in *Figure 4A*. An adapted MRI protocol (*Rosenzweig et al., 2018*) was used to assess T1ρ-weighted MRI signal that directly correlates with proteoglycan content (*Rosenzweig et al., 2018*; *Mulligan, 2015*). The method used to assess uniform NP regions of interest is shown in (*Figure 2—figure supplement 1B-D*). The same NP regions of interest were scanned in the same orientation pre- and post-treatment. Increased T1ρ values were found in RG-7112 and o-Vanillin-treated discs while vehicle-treated discs displayed decreased T1ρ values. (*Figure 4B* (a-d)) Quantification of pre- versus post- treatment intensity values showed a non-significant decrease of 13.2 ± 4.7% (p=0.058) in vehicle-treated control IVDs. In RG-7112 treated discs, there was a significant 6.8 ± 1.5% (p=0.024) increase in the T1ρ value post-treatment. Discs that were treated with o-Vanillin, displayed a significant increase of 11.1 ± 1.2% (p=0.001) in T1ρ values following treatment (*Figure 4C*). The red dye safranin-O, binds negatively charged molecules, which are primarily represented by proteoglycan in the NP. Histological evaluation post-treatment using Safranin-O/fast green staining showed strong (intense red) staining in treated discs (*Figure 4D*). Finally, p16$^{Ink4a}$ and Ki-67 immunohistochemistry were performed to verify senolytic activity of the two compounds (*Figure 4E*). Indeed, immunohistochemical assessment of p16$^{Ink4a}$ showed a significant decrease in the number of senescent cells of 17.38 ± 1.6% (p=0.0017) in RG-7112 and 22.65 ± 3.6% (p=0.008) in o-Vanillin-treated IVDs compared with vehicle-treated control discs (*Figure 4F*). Quantification of Ki-67 staining showed a non-significant increase in both RG-7112 and o-Vanillin 3.05 ± 3.4% (p=0.4) treated IVDs (*Figure 4G*). The data suggest that both senolytics can reach and kill naturally occurring senescent human IVDs cells situated in their native environment and at the same time promote tissue repair and regeneration.

## RG-7112 and o-Vanillin decreased SASP factor release from intact human IVDs

Culture media from treated and untreated discs (in *Figure 4*) was analyzed using cytokine arrays and were compared to their respective pre-treatment media. Interestingly, a single injection with RG-7112 strongly decreased secretion of several proinflammatory cytokines and chemokines including IL-7 (−60.63% (p=0.01)), IL-6 (−59.39% (p=0.14)), CXCL1 (−36.72% (p=0.26)), GRO-abg (−32.84% (p=0.21)) and CCL24 (−30.57% (p=0.15)). o-Vanillin also strongly decreased IL-7 (−83.37% (p=0.09)), CXCL1 (−65.45% (p=0.12)), IL-6 (−58.68% (p=0.10)), CXCL6 (−52.47% (p=0.009)), IGFBP-2 (−44.51% (p=0.15)), GRO-abg (−41.83% (p=0.03)) and CCL2 (−39.91% (p=0.05)). Both compounds significantly decreased IL-8 by 18.72% (p=0.004) for RG-7112 and by 11.75% (p=0.04) for o-Vanillin. We also detected a moderate increase of CXCL9, CCL22, NT3 and EGF for o-Vanillin and only in CCL2 and OPN for RG-7112-treated media (*Figure 5A*). The level of the SASP factors released from vehicle-treated discs showed an increase in IL-6 (530.85% (p=0.15)), CXCL6 (196.89% (p=0.07)), CXCL1 (99.83% (p=0.18)), OPN (92.36% (p=0.01)), CCL2 (53.37% (p=0.18)), IGFBP-2 (39.26% (p=0.13)), IL-7 (23.52% (p=0.53), GRO-abg, (14.25% (p=0.91)), CCL24 (9.64% (p=0.97)), EGF (3.18% (p=0.85)), TNSF-14 (2.09% (p=0.66)), IL-8 (1.61% (p=0.82)) and a decrease in CXCL9 (−35.23% (p=0.26)), CCL22 (−11.95% (p=0.56)) and NT3 (−13.46% (p=0.25))(*Figure 5B*). The complete set of factors from all groups are presented in *Figure 5—figure supplement 1 (A-C)*.

Luminex quantification of the same 19 selected SASP factors that were analyzed in pellet cultures showed an overall decrease in inflammatory mediators in culture media from both RG-7112 or

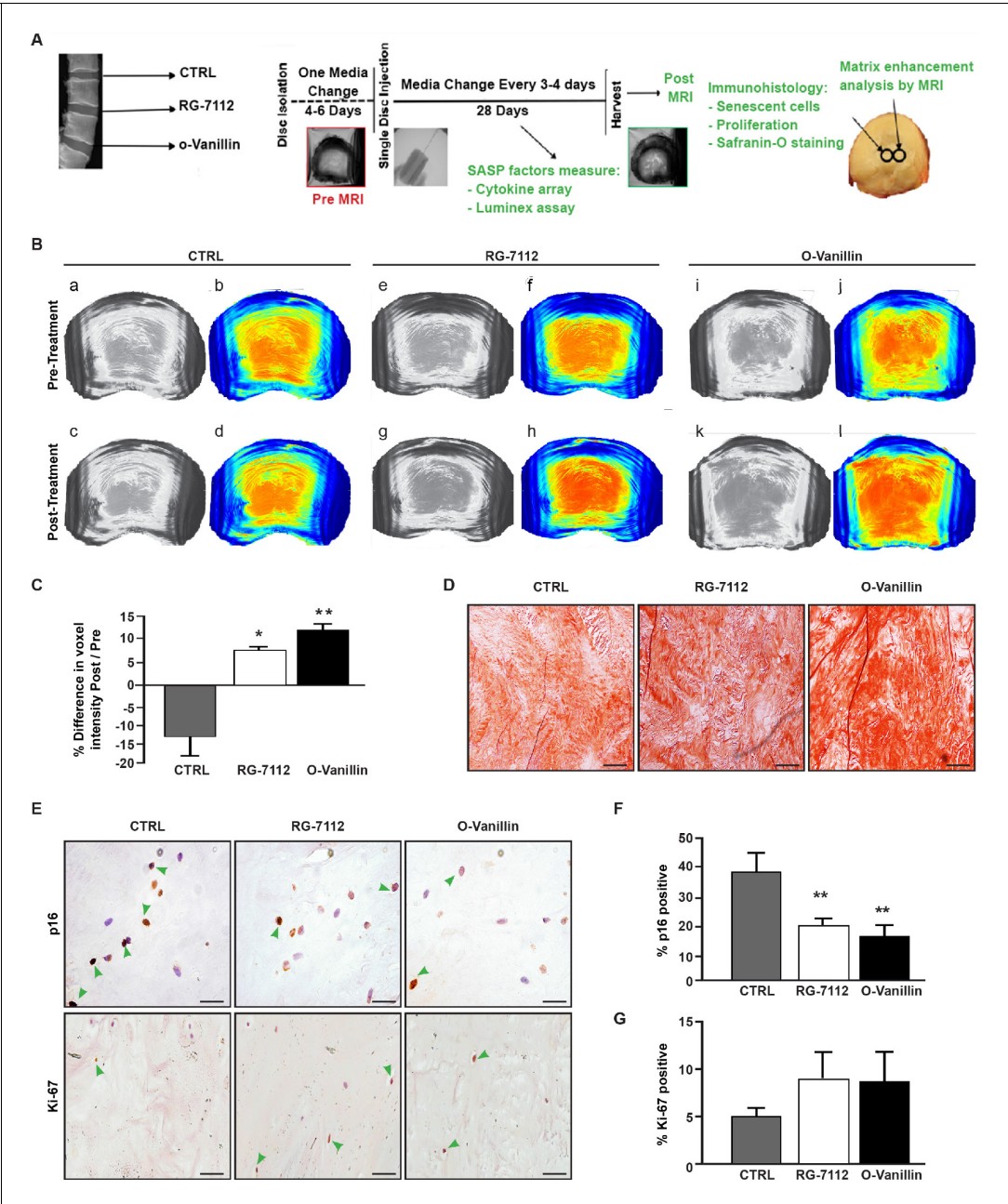

**Figure 4.** RG-7112 and o-Vanillin effects in ex vivo human IVD culture. (**A**) Schematic of the ex vivo organ culture experiment. Lumbar spines from organ donors were assessed radiographically for signs of degeneration. Three discs per experiment were isolated from the same spine, cultured for 4–6 days then scanned with MRI and injected with vehicle, o-Vanillin or RG-7112. Discs were then cultured for an additional 28 days, with media changes every 3–4 days. The discs were scanned by MRI again at day 28. Media and tissues were used for SASP factor release and histology respectively. (**B**) Representative images of mid-axial T1ρ slices pre-treatment (**a–b, e–f, i–j**) and the same location post treatment (**c–d, g–h, k–l**) with vehicle (CTRL), RG-7112 (5 µM) or o-Vanillin (100 µM). The heatmap correlates the red color with the highest and the blue color with the lowest T1ρ values. (**C**) Quantification for NP regions with the graph showing percentage change in T1ρ values post- compared to the pre-treatment scans. (**D**) Representative safranin O/fast green staining of histological sections. (**E**) Representative images of disc sections stained with antibodies against p16$^{INK4a}$ and Ki-67. Quantification of (**F**) p16$^{INK4a}$ and (**G**) Ki-67 expression. Scale bars = 150 µm in 4D, 25 µm in 4E (p16 $^{INK4a}$) and 50 µm in 4E (ki-67); Error bars represent mean ± SEM, Statistical significance was assessed by two-tailed Student's t-test to compare pre and post disc groups (**C**) and by repeated measures Analysis of Variance (ANOVA) with Turkey's post hoc test for multiple pairwise comparison in (**F and G**). *Indicates p<0.05 and **indicates p<0.01, n = 4 for each condition. The tissues were from degenerating IVDs as indicated in *Tables 2* and *3*.

The online version of this article includes the following source data for figure 4:

**Source data 1.** Excel file containing the complete list of donors, slices and their respective voxel intensities, p16$^{Ink4a}$ and ki-67-positive cells in control and treated discs.

o-Vanillin-treated discs compared with that of vehicle (*Figure 5C*). All factors decreased in RG-7112 and o-Vanillin-treated discs (*Figure 5D* and *Figure 5—figure supplement 1D*). Six factors in RG-7112 and five in o-Vanillin-treated discs were significantly decreased in the post-treatment media compared with their respective pre-treatment media. For RG-7112, these proteins were TNF-α (mean 12.4 pg/ml in pre, 6.8 pg/ml in post, p=0.03), CCL11 (mean 13.1 pg/ml in pre, 7.8 pg/ml in post, p=0.009), CCL24 (mean 201.7 pg/ml in pre, 101.1 pg/ml in post, p=0.04), CXCL1 (mean 261.8 pg/ml in pre, 148.2 pg/ml in post, p=0.001), CXCL10 (mean 197.1 pg/ml in pre, 158.7 pg/ml in post, p=0.04) and Angiogenin (mean 51.7 pg/ml in pre, 37.6 pg/ml in post, p=0.02). o-Vanillin significantly reduced the levels of INF-γ (mean 1.6 pg/ml in pre, 1.1 pg/ml in post, p=0.04), CCL24 (mean 232.7 pg/ml in pre, 73.7 pg/ml in post, p=0.02), CXCL1 (mean 313.9 pg/ml in pre, 130.2 pg/ml in post, p=0.01), Angiogenin (mean 47.4 pg/ml in pre, 35.5 pg/ml in post, p=0.02) and VEGF-A (mean 24802.1 pg/ml in pre, 20905.5 pg/ml in post, p=0.04) (*Figure 5D*). Finally, we found a significant increase in the release of four factors in vehicle treated discs: INF-γ (mean 1 pg/ml in pre, 1.45 pg/ml in post, p=0.02), CCL11 (mean 7 pg/ml in pre, 19.2 pg/ml in post, p=0.04), CXCL1 (mean 104.3 pg/ml in pre, 476.9 pg/ml in post, p=0.03) and CXCL9 (mean 60.4 pg/ml in pre, 162.4 pg/ml in post, p=0.01).

## Discussion

We have previously demonstrated that curcumin and its metabolite o-Vanillin have senolytic activity toward senescent human IVD cells (*Cherif et al., 2019*). Treatment with curcumin and o-Vanillin reduced SASP factors released and enhanced matrix synthesis in a pellet culture model (*Cherif et al., 2019*). In this study, we aimed to compare the effects of a synthetic pure senolytic compound with a natural anti-inflammatory and senolytic compound, to determine a potential enhanced therapeutic effect with the latter. We used o-Vanillin instead of curcumin, as o-Vanillin has higher specificity and better bioavailability. For a synthetic and pure senolytic we chose RG-7112, a drug with documented senotherapeutic effects in fibroblasts (*Weber, 2010*; *Laberge et al., 2018*). Cells from degenerate discs or intact degenerate discs that correspond with the tissue targeted for treatment were used. Cells from degenrating tissue was used as it would be difficult to evaluate an effect of removing senescent cells in non-degenerate discs since they have have very few senescent cells. Here, we demonstrated that RG-7112 has a potent senotherapeutic and a strong proliferative effect on human IVD cells in vitro. Metabolic activity and extracellular matrix production were also increased in treated cultures. We verified that treatment specifically targets senescent cells by activating the caspase-3 apoptotic pathway. This is similar to the effect of UBX0101, an analogue of RG-7112 that triggers apoptosis of senescent chondrocytes in a murine osteoarthritis model (*Jeon et al., 2017*). Currently, phase I and II clinical trials are in progress to assess safety, tolerability and clinical effects of single dose (NCT03513016 and NCT04129944) and both single and repeated doses (NCT04229225) of intra-articular administration of UBX0101 in patients with moderate to severe painful knee osteoarthritis (*van Deursen, 2014*; *Vassilev et al., 2004*; *Vu et al., 2013*).

Cellular senescence can be induced by replicative senescence or stress-induced premature senescence (*Wang et al., 2016*). The pathway and resulting changes in the microenvironment surrounding the cells depends on inducing factor (*Frolov and Dyson, 2004*). In this study, we found gene expression modulation for a number of inflammatory and cell cycle genes following treatment with the two senolytics, o-Vanillin and RG-7112. We evaluated differential gene expression in 96 pre-selected genes to determine the mechanisms by which the compounds mediate their effects. Our data demonstrate that 91 of the genes were expressed at a detectable level and a significant effect was found on 50% of the expressed genes, supporting the senolytic activity of the two compounds in vitro. RG-7112 decreased gene expression of CDK1A and MDM2 and increased expression of the E2F1, RB factors, MAPK-14, CDK2D, CCNB1 and CDC25c. o-Vanillin modulated gene expression of 40 genes including upregulation of cell cycle genes such as CDK6, CDK2C, CDK2D, and CDC25c while expression levels of CDK2A, Cyclin A2, Cyclin D1 and CCNB1 were decreased.

Collectively, the data support the mechanistic action of RG-7112 to stabilize p53 and p21 by attenuating the MDM2-p53 interaction (*Weber, 2010*; *Tovar et al., 2013*; *Henley and Dick, 2012*; *Che et al., 2020*). Also, it confirms regulation of the RB–E2F1 pathway that releases E2F1 and activates genes involved in cell cycle regulation, DNA synthesis, and cell proliferation (*Andreeff et al., 2016*; *Xu et al., 2018*). Another example of a senolytic drug that interferes with the E3 ubiquitin

ligase-MDM2-p53 mechanism is UBX0101, which triggers apoptosis of senescent cells in articular cartilage and synovium in a murine osteoarthritis model (*Jeon et al., 2017*). o-Vanillin treatment affected multiple pathways suggesting that o-Vanillin eliminates senescent cells both by apoptosis and non-apoptotic means. Recently, p16 deletion in NP cells from mouse discs with induced degeneration was shown to also upregulate the expression of cyclin-dependent kinases 4/6, phosphorylated retinoblastoma protein, and transcription factor E2F$_{1/2}$ (*Che et al., 2020*).

Clinical trials with RG-7112 for cancer treatment were limited by the high incidence of hematological toxicities (*Ray-Coquard et al., 2012*; *Jung et al., 2008*). Here, we used a lower concentration (5 µM) and short exposure time of RG-7112, in contrast to the high and toxic doses used in cancer therapy (20–1400 mg/m$^2$ administered daily for 10 days). This may prevent these side effects in patients treated for IVD degeneration. This hypothesis was verified by the clinical study of the RG-7112 analogue UBX0101 in patients diagnosed with painful osteoarthritis of the knee. An administered dose of 4 mg was safe and well-tolerated, and it improved pain scores, reduced SASP factors and disease-related biomarkers after a single dose (*Weber, 2010*; *Laberge et al., 2018*). The activation of multiple pathways following o-Vanillin treatment and within the selected set of 91 genes implies that diverse biological processes are affected, which was also reported for other natural senolytics like Quercetin, Fisetin, and Dasatinib (*Pezet and McMahon, 2006*; *Wang et al., 2008*). Although it is difficult to determine how these natural senolytics eliminate senescent cells and drive their therapeutic effects, the activation of several mechanisms increases their capacity to target the heterogeneous cellular states acquired by senescent cells after the initial growth arrest (*Frolov and Dyson, 2004*).

We further sought to determine if the two senolytics had apparent effects on the SASP secretome. Conditioned media of pellet cultures was analyzed by cytokine array following treatment. Although donor variation was observed, notable changes between treatment groups were found indicating an overall decrease in the majority of the SASP factors analyzed. Similar decrease in the SASP factors was observed following p16 deletion in NP cells in mice with induced disc degeneration (*Yousefzadeh et al., 2018*). An antiinflammatory effect was expected for o-Vanillin from previous studies (*Cherif et al., 2019*; *Santosh Kumar et al., 2002*; *Oliveira et al., 2014*; *Shah et al., 2019*; *Marton et al., 2016*). However, our study documents, for the first time, antiinflammatory properties of RG-7112 in human IVD cells. We selected 19 factors from the initial screening associated with SASP and measured their concentrations using a Luminex immunoassay. In general, concentrations were decreased in response to treatment. RG-7112 significantly reduced the concentrations of IFN-γ, IL-6 and CCL24 while o-Vanillin in addition reduced factors IL-8, CCL7, CXCL1, CXCL5, CXCL9, CXCL10 and VEGF-A. These factors are known for their implication in painful IVD degeneration (*Acosta and Gil, 2009*; *Acosta et al., 2008*; *Phillips et al., 2013*) and for promoting senescence of surrounding cells (*Coppé et al., 2010*; *Adams et al., 2015*). For example, IL-8 and chemokines binding to the C-X-C motif chemokine receptors is needed for the establishment and maintenance of senescence. CCL7 expression has been reported to be concordant with IVD degeneration (*Binch et al., 2014*) and it has previously been described as a SASP factor involved in IVD cell senescence (*Burke et al., 2002*). Finally, the decrease of VEGF-A indicates a possible role of o-Vanillin to block the mechanism of neovascularization. VEGF and its receptors have been proposed to be closely correlated with inflammation, chronic back pain and accelerated IVD degeneration (*Freemont et al., 1997*; *Lu et al., 2013*; *Peng et al., 2006*; *Daly et al., 2016*; *Gruber et al., 2009*; *Thompson et al., 1991*). These results consolidate the senolytic effect of the two compounds and suggest a potential role in reducing degenerative factors in human discs with o-Vanillin having a possible stronger and broader antiinflammatory effect.

Animal models that replicate human disc pathology are limited due to the differences in anatomy, disc size, cell type, and loading (*Jin et al., 2018*). Moreover, notochord cells are retained longer in the majority of animal species (*Grezella et al., 2018*), further increasing the difference between humans and animals. Thus, we previously developed and validated an ex vivo intact human disc culture system to study the potential for biologic repair and regeneration (*Gawri et al., 2011*). Matrix differences were quantified pre- and post-treatment by T1ρ weighted MRI that directly correlates with proteoglycan content in the IVDs (*Rosenzweig et al., 2016*; *Mulligan, 2015*). When comparing the intensity in RG-7112 and o-Vanillin-treated IVDs, we observed an increased intensity post-treatment, while a decrease was observed in vehicle-treated control. This suggests that treatment with senolytics could increase proteoglycan content in human patients. It also validates our results from the pellet cultures where the proteoglycan content also increased significantly in the treated

cultures. The improvement of T1ρ MRI values was validated by the consistent high levels of proteoglycan content in the treated IVDs. We further verified that the improved matrix was linked to removal of senescent cells. Indeed, both compounds significantly reduced the number of p16$^{Ink4a}$-positive cells in treated discs. Moreover, we observed a slight but non-significant increase in proliferating cells. These findings correlate with the negative impact previously reported between the number of senescent cells and cell proliferation during IVD degeneration (*Hwang et al., 2018*). Furthermore, culture media from control discs, showed no change in cytokine release whereas both compounds decreased levels of IL-6, IL-7, IL-8, CXCL1, CXCL6, CCL2, CCL22, GROa/b/g, IGFBP-2, TNSF-14 and OPN. Moreover, we observed a significant increase of EGF in the media from o-Vanillin treated discs. The positive effects of EGF on proteoglycan synthesis were first reported by Thompson et al.; these effects were more pronounced in the NP (*Zhu et al., 2017*). Together the results demonstrate an overall decrease in SASP factors following treatment. The difference in SASP compounds affected by the two drugs could be explained by the difference in selectivity and the specificity of the drugs (*Zhu et al., 2017*; *Zhu et al., 2016*; *Wilke et al., 2006*; *Thompson et al., 1990*; *Gawri et al., 2014*).

Finally, we compared the SASP factors release in pellet and disc culture media. The common SASP factors that were downregulated include INF-γ, IL-6, CCL24, CXCL1, CXCL10 and Angiogenin. Interestingly, o-Vanillin also significantly reduced the levels of IL8, CCL7, CXCL5, CXCL9 and VEGF-A. This could be due to the effect on the neighboring non-senescent cells. Consistent with prior literature, this decrease highlights the antioxidant and antiinflammatory properties of o-Vanillin (*Cherif et al., 2019*; *Santosh Kumar et al., 2002*; *Oliveira et al., 2014*; *Shah et al., 2019*; *Marton et al., 2016*). Also, these decreases have been reported in several other natural compounds such as quercetin, fisetin, and piperlongumine (*Wang et al., 2008*; *Thompson et al., 1990*).

Our findings suggest that both o-Vanillin and RG-7112 have the potential to be translated to treatment of painful IVD degeneration. As well, this study can be used as a base for the development of additional senolytic agents for lower back pain.

# Materials and methods

### Key resources table

| Reagent type (species) or resource | Designation | Source or reference | Identifiers | Additional information |
|---|---|---|---|---|
| Antibody | Rabbit polyclonal anti-caspase-3 | Sigma-Aldrich | C8487 RRID:AB_476884 | IF (1:500) IHC-Fr (1:500) |
| Antibody | Rabbit polyclonal anti-Ki-67 | Novus Biologicals | NB500-170 RRID:AB_10001977 | IF (1:300) IHC-Fr (1:500) IHC-P (1:2000) |
| Antibody | Mouse monoclonal Anti- p16$^{Ink4a}$ | Abcam | ab54210 RRID:AB_2059963 | IHC-P (1:500) |
| Antibody | Polyclonal Alexa Fluor 488 (Donkey anti-Rabbit) | Invitrogen Thermo Fisher | A-21202 RRID:AB_141607 | IF (1:1000) |
| Antibody | Polyclonal Alexa Fluor 555 (Donkey anti-Mouse) | Invitrogen Thermo Fisher | A-31570 RRID:AB_2536180 | IF (1:1000) |
| Antibody Array | Human Cytokine Antibody Array C5 | RayBiotech, Inc | AAH-CYT-5–2 RRID:AB_10185250 | |
| Chemical compound, drug | o-Vanillin | Sigma-Aldrich | 120804 | |
| Chemical compound, drug | RG-7112 | Selleck Chemicals | S7030 | |
| Chemical compound, drug | DMSO | Sigma-Aldrich | D8418 | |
| Commercial Kit | Apoptosis detection | Abcam | ab176749 RRID:SCR_018379 | |
| Commercial Kit | Caspase 3/7 activity | AAT Bioquest 13503 | Amplite Fluorimetric Catalog # 13503 | |

*Continued on next page*

*Continued*

| Reagent type (species) or resource | Designation | Source or reference | Identifiers | Additional information |
|---|---|---|---|---|
| Commercial Kit | qScript cDNA Synthesis Kit | Quanta Biosciences | 95047–500 | |
| Commercial Kit | Goat anti-Mouse and Rabbit specific HRP/DAB Detection IHC Kit | Abcam | ab64264 | |
| Commercial Kit | Mouse monoclonal Anti- p16$^{Ink4a}$ | Roche, Ventana Laboratories | CINtec Plus (9531) RRID:SCR_018380 | Prediluted |
| Commercial Kit | Mycoplasma PCR Detection Kit | ZmTech Scientific | M209001 | |
| Commercial Assay | Custom Taqman (R) Gene Expression 96-Well Fast Plates | Thermo Fisher | 4413255 | See *Supplementary file 1* for each gene product ID |
| Commercial Assay | Luminex immunoassay 19-PLEX plate - Human Cytokine PROCARTAPLEX | Life Technologies | MXMFXHX | |
| Software | ImageJ | ImageJ http://imagej.nih.gov/ij/ | RRID:SCR_003070 | |
| Software | GraphPad Prism 8 | GraphPad Prism https://www.graphpad.com | RRID:SCR_015807 | |
| Software | Ingenuity Pathway Analysis (IPA) | Ingenuity Systems | RRID:SCR_008653 | |
| Software | R package ggplot2 | *Wickham, 2016* | ISBN 978-3-319-24277-4 | |
| Software | MIPAV | NIH Center for Information Technology | RRID:SCR_007371 | |
| Software | Image Quant TL array analysis | GE Healthcare | RRID:SCR_018374 | |
| Software | FlowCytomix Pro2.2.1 | eBioscience | RRID:SCR_018375 | |
| Software | AxioVision LE64 | Zeiss | RRID:SCR_002677 | |
| Other | DMMB | Sigma-Aldrich | 341088 | |
| Other | Alamar Blue reagent | Thermo Fisher | DAL1025 | |

*Continued on next page*

**Table 1.** Donors demographics.

| Donor ID | Age | Sex | Cause of death |
|---|---|---|---|
| 1 | 53 | F | Anoxia |
| 2 | 53 | M | bone; cerebral aneurysm rupture |
| 3 | 76 | F | Anoxia |
| 4 | 52 | M | Ischemic cerebral vascular accident |
| 5 | 17 | M | Brain death |
| 6 | 73 | F | Cerebrovascular hemorrhage |
| 7 | 68 | F | Cerebral hemorrhage |
| 8 | 39 | M | Gunshot to the neck |
| 9 | 66 | F | Unknown |
| 10 | 67 | M | Cardiac arrest |
| 11 | 69 | M | Cerebral hemorrhage |
| 12 | 61 | M | Cardiac arrest |
| 13 | 40 | F | Brain hemorrhage |
| 14 | 53 | F | Medical suicide |
| 15 | 28 | M | Suicide (Hang) |
| 16 | 35 | M | Anoxia |

**Table 2.** Complete donors list per assays and results.

| Donor ID | Discs | Thompson grade and Degeneration | Cell type | Type of culture | Type of assays | Results |
|---|---|---|---|---|---|---|
| 6 | L1-L5 | 3 - (D) | NP & AF | Pellet | IHC Metabolic activity | *Figure 1 (A–D)* Donor # 6, 7, 10–15 |
| 7 | L2-L5 | 4 - (D) | | | | *Figure 1—figure supplement 1(A)* |
| 8 | L3-S1 | 3 - (D) | | | | Donor # 10–15 |
| 10 | L4-L5 | 4 - (D) | | | | *Figure 1—figure supplement 1(G)* |
| 11 | L5-S1 | 3 - (D) | | | | Donor # 10–15 |
| 12 | L5-S1 | 3 - (D) | | | | |
| 13 | L4-S1 | 3 - (D) | | | | |
| 14 | L5-S1 L1-L2 | 3 - (D) | | | | |
| 15 | L5-S1 | 5 - (D) | | | | |
| 11 | L5-S1 | 3 - (D) | NP | Pellet | RT-qPCR Cytokine array Luminex | *Figure 2 (A–E)* *Figure 3 (A–E)* |
| 12 | L5-S1 | 3 - (D) | | | | *Figure 2—figure supplement 1(A)* |
| 13 | L4-S1 | 3 - (D) | | | | *Figure 3—figure supplement 1(A-C)* |
| 14 | L5-S1 L1-L2 | 3 - (D) | | | | All Donor # 11–15 |
| 15 | L5-S1 | 5 - (D) | | | | |
| 9 | T11-L2 | *Table 3* | Intact disc | Ex vivo | MRI Luminex IHC | *Figure 4 (A–F)* *Figure 5 (A–D)* |
| 11 | L1-L4 | *Table 3* | | | | *Figure 5—figure supplement 1(A-D)* |
| 13 | L1-L4 | *Table 3* | | | | Donor # 9, 11, 13 and 14 |
| 14 | L2-L5 | *Table 3* | | | | |
| 1 | L1-S1 | 4 - (D) | NP | Monolayer | ICC Metabolic activity Caspase 3/7 activity | *Figure 1—figure supplement 1(B, C and D)* |
| 2 | L4-S1 | 3 - (D) | | | | Donor # 1–3 |
| 3 | L2-S1 | 4 - (D) | | | | *Figure 1—figure supplement 1(E and F)* |
| 4 | L1-L5 | 1 - (ND) | | | | Donor # 1–3 and 11 (D) |
| 5 | T11-S1 | 1 - (ND) | | | | Donor # 4, 5, 10 and 11 (ND) |
| 10 | T12-L1 | 1 - (ND) | | | | |
| 11 | T12-L1 L5-S1 | 1 - (ND) 3 - (D) | | | | |
| 15 | T11-L5 L5-S1 | 1 - (ND) 5 - (D) | Intact disc | NP Area and position measure | | *Figure 2—figure supplement 1(B-C)* Donor # 15 and 16 |
| 16 | L1-S1 | 1 - (ND) | | | | |

*Continued*

| Reagent type (species) or resource | Designation | Source or reference | Identifiers | Additional information |
|---|---|---|---|---|
| Other | DAPI stain | Invitrogen | D1306 | (1 µg/mL) |
| Other | Safranin-O | Sigma-Aldrich | S2255 | |
| Other | TaqMan Fast Universal PCR Master Mix (2×) | Thermo Fisher | 4366073 | |
| Cell line | Primary Human Nucleus Pulposus (NP) and Annulus Fibrosis (AF) | Primary Cells | Human Biological samples. See *Table 1* for demographics. | Cell line maintained in Haglund Lab. |

## Human disc collection

Human lumbar IVDs were harvested from organ donors through a collaboration with Transplant Quebec. All procedures are approved by and performed in accordance with the ethical review board

**Table 3.** Characteristics of discs injected with each treatment.

| | Donor ID | 9 | 11 | 13 | 14 |
|---|---|---|---|---|---|
| | **Age** | 66 | 69 | 40 | 53 |
| | Sex | F | M | F | F |
| CTRL | Disc Level | T12/L1 | L1/2 | L1/2 | L2/3 |
| | Grade | 2 | 2 | 2 | 2 |
| | Disc Height (cm) | 0.84 | 0.92 | 0.92 | 0.88 |
| O-Vanillin | Disc Level | L1/2 | L2/3 | L3/4 | L4/5 |
| | Grade | 2 | 2 | 2 | 2 |
| | Disc Height (cm) | 0.92 | 0.89 | 0.99 | 0.88 |
| RG-7112 | Disc Level | T11/T12 | L3/4 | L2/3 | L3/4 |
| | Grade | 2 | 2 | 2 | 2 |
| | Disc Height (cm) | 0.82 | 0.9 | 0.93 | 0.99 |

at McGill University (IRB#s A04-M53-08B). Familial consent was obtained for each subject. *Table 1* provides an overview of donor demographics and *Table 2* provides detailed description of the used discs. Lumbar spinal columns were removed from organ donors, they were imaged radiographically and visually, and signs of degeneration were noted (*Page et al., 1993*; *Mort and Roughley, 2007*). Discs were then dissected from the spinal column and used for cell and organ cultures. Nucleus pulposus (NP) and annulus fibrosis (AF) cells were isolated separately as described previously (*Cherif et al., 2019*; *Kirkland and Tchkonia, 2017*). All cultures were mycoplasma free as verified by Mycoplasma PCR Detection Kit (ZmTech Scientific).

## Alamar blue metabolic activity in monolayer cell culture

10,000 cells were seeded per well in 96-well tissue culture plates for 12 hr. Then, cells were incubated in the presence or absence of RG-7112 (5 µM) for 6 h. Metabolic activity was evaluated by the Alamar blue assay as previously described (*Livak and Schmittgen, 2001*). We measured the metabolic activity of treated and untreated NP cells cultured in monolayer or in pellet from degenerate and non-degenerate discs to verify a non cytotoxic window of RG-7112. Results are presented as a percentage of metabolic activity compared to their control. Experiments were performed (3–6) times in triplicate wells for each compound and concentration.

## Pellet cell culture and immunohistochemistry

300,000 NP cells/tube were collected by centrifugation at $500 \times g$ for 5 min. Pellets were incubated in 1 mL DMEM (2.25 g/L glucose, 5% FBS, ascorbic acid (5 µM) (Sigma-Aldrich, Oakville, ON, Canada)) at 37°C and 5% $CO_2$. Culture media was changed every 3 days and collected for analysis. After 4 days, the pellet culture is stabilized and a single dose of the senolytics or vehicle was added to the culture media. Metabolic activity measures on pellet culture media from degenerate NP and AF cells, treated or not with RG-7112, was performed to evaluate a potential cytotoxic effect of RG-7112 by comparing, in the same pellet culture, the effect of the treatment (at day 21) to the control before treatment (at day 4). RNA was collected in TRIzol (Thermo Fisher Scientific) for gene expression experiments (as described in *Krock et al., 2014*). For immunohistochemistry, pellets were fixed in 4% paraformaldehyde and cryopreserved, 5 µm sections were prepared and stained overnight at 4°C for p16$^{Ink4a}$ and 1 hr at room temperature for Ki-67 and caspase-3 primary antibodies as described (*Cherif et al., 2019*). Coverslips were mounted using Aqua Polymount, and bright-field images were visualized. Ten fields, randomly distributed across the well, were analyzed, and the number of positive (brown stained) and total cells were counted to calculate the percentage of senescent, proliferative and apoptotic cells. Senescence, Apoptosis and proliferation assessments were performed on cells pellet cultured for 21 days by comparing, in the same subject, the senolytics-treated pellet to their respective controls.

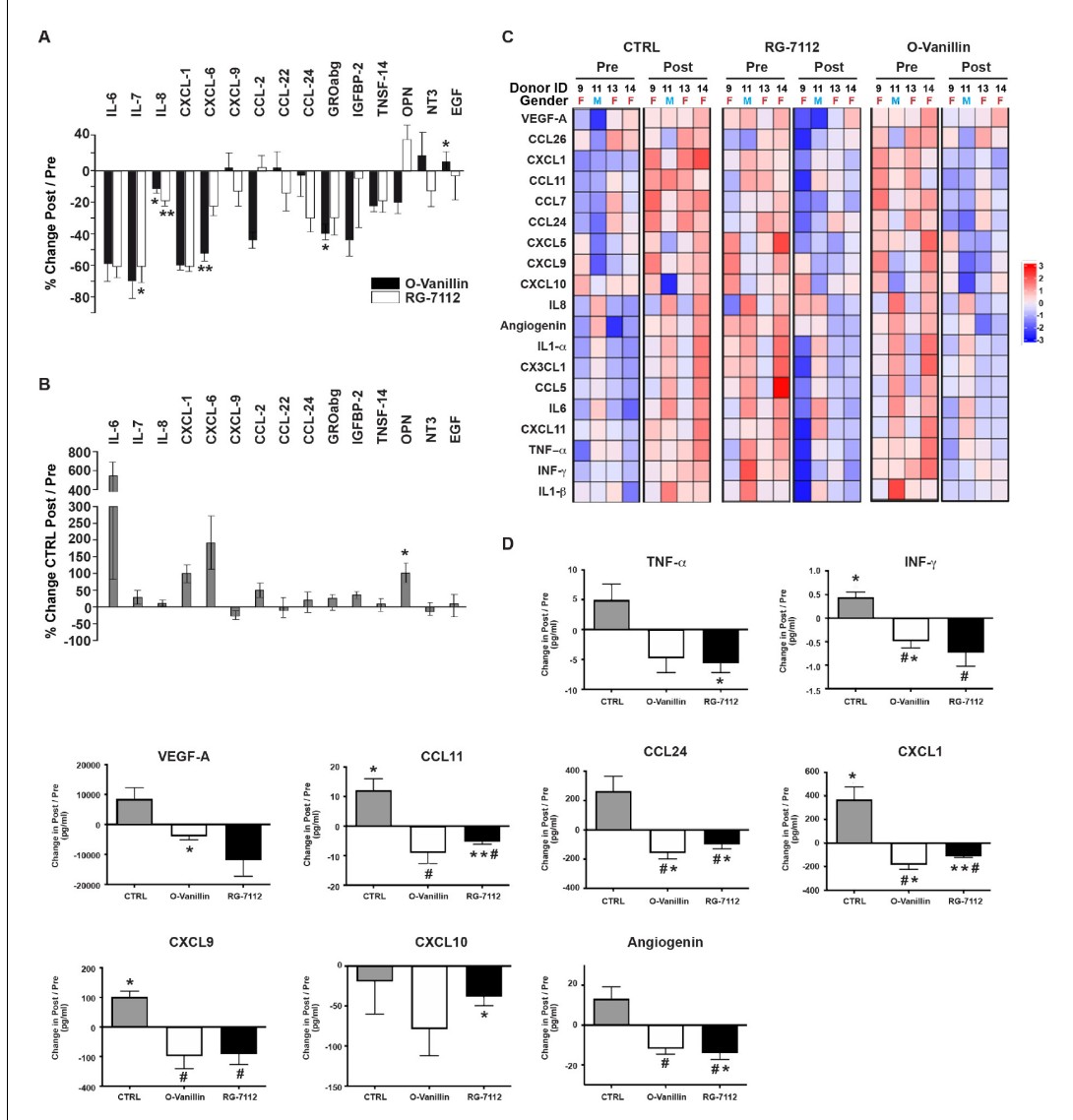

**Figure 5.** Profile of SASP factors released from ex vivo human IVDs cultures following senolytic treatment. Culture media was analyzed by a RayBio Human Cytokine Array. Relative mean densitometry units of the 80 factors were normalized to pre-treatment media of the same IVD. The percentage change (post/pre) of the 15 most affected factors are shown for RG-7112 and o-Vanillin (**A**) and vehicle (**B**) treated discs. (**C**) Heatmap displaying quantification of 19 selected cytokines (19-plex Luminex array). Each column represents one individual and each row represent expression of a single protein. Data shown are log2 (fold change) in pre- or post-treated disc media relative to their respective expression level average. (**D**) Nine analytes (INF-γ, TNF-α, CCL11, CCL24, CXCL1, CXCL9, CXCL10, Angiogenin and VEGF) displayed statistically significant differences when measured in post compared to pretreated disc media. Error bars represent mean ± SEM. Statistical significance when comparing pre and post discs groups was assessed by two-tailed Student's t-test (**A, B and D**): * Indicates p<0.05 and **indicates p<0.01. Data was analyzed by repeated measures Analysis of Variance (ANOVA) with Turkey's post hoc test for multiple pairwise comparison (**D**) where # Indicates significant difference (p<0.05) between treated and untreated groups, (n = 4). The analyzed media were collected from same donors used in *Figure 4*.

The online version of this article includes the following source data and figure supplement(s) for figure 5:

**Source data 1.** Excel file containing the meausres of the significantly changed SASP factors measured using cytokine array and Luminex analysis in vehicle, RG-7112 and o-Vanilin disc media before and after the injection.
**Figure supplement 1.** Complete profiling and measure of growth factors and cytokines in disc cultures media.
**Figure supplement 1—source data 1.** Excel file containing the complete list of SASP factors measured by cytokine arrays and the non-significantly regulated factors measured by Luminex assay in the pre and post media of injected discs.

## Monolayer cell culture and immunofluorescence

Isolated cells were expanded to Passage 1 (P1) in monolayer cultures. P1 cells were then seeded at 20,000 or 10,000 cells per well in eight-well chamber slides (Nunc Lab-Tek II Chamber Slide System) and 96- well flat clear bottom black microplates (Corning, NY) respectively. Cells were serum-starved in DMEM with ITS (1X) (Thermo Fisher, Waltham, MA) for 2 hr prior to treatment with 5 µM RG-7112 (Selleck Chemicals, TX), 100 µM o-Vanillin (Sigma-Aldrich, Oakville, ON, Canada) or vehicle (DMSO (0.01%, (Sigma-Aldrich, Oakville, ON, Canada) for 6 hr. Immunocytochemistry was performed as previously described (*Cherif et al., 2019*). Apoptosis was detected using a commercial kit (ab176749, Abcam, Cambridge, MA) according to the manufacturer's instructions. Photomicrographs were acquired with a fluorescent Olympus BX51 microscope equipped with an Olympus DP71 digital camera (Olympus, Tokyo, Japan).

## Caspase 3/7 activity in monolayer cell culture

Caspase 3/7 activity of treated and untreated NP cells from degenerate and non-degenerate IVDs was measured using the Amplite Fluorimetric Caspase 3/7 Assay Kit (AAT Bioquest, Sunnyvale, CA) according to the manufacturer's protocol. Cells were incubated with the caspase 3/7 assay solution, which contained caspase substrate (Z-DEVD-R110), at room temperature for 1 hr in the dark. Fluorescence intensity was then measured at 490 nm excitation and 525 nm emission. The results are expressed as a percentage of the mean of the control group (set at 100%). Each experiment was performed in triplicate and carried out three times from each round of cell isolation.

## DMMB in pellet cell culture

Sulphated glycosaminoglycans (GAGs) were quantified using the DMMB assay in the media of degenerate NP and AF pellets with or without RG-7112 treatment, performed as previously described (*Wickham, 2016*). Chondroitin sulfate was used to generate the standard curve. Conditioned media samples from days 7, 10, 14 and 21 were evaluated separately in triplicate into clear 96-well plates (Costar, Corning, NY). sGAG release in media was normalized to the sGAG concentration in media at day 0 and then normalized to the untreated group.

## Quantitative real-time PCR in pellet cell culture

Following treatment, RNA was extracted using the TRIzol chloroform extraction method, as previously described (*Krock et al., 2014*). Briefly, 500 ng of RNA was reverse transcribed using a qScript cDNA Synthesis Kit (Quanta Biosciences, Beverly, MA) with an Applied Biosystems Verti Thermocycler (Thermo Fisher, Waltham, MA). RT-qPCR was performed using an Applied Biosystems StepOne-Plus machine with TaqMan Fast Universal PCR Master Mix (2×) and Custom TaqMan Array 96-Well Fast Plates (Thermo Fisher, Waltham, MA). The evaluated genes are senescent and apoptotic genes included in the Human TaqMan Array, for Human Cellular Senescence (Thermofisher scientific, Array ID: RPU62T7, Catalog number: 4413255). The Array Plate was customized to include 12 additional recently identified senescence and anti-apoptotic pathway genes[84]. The 96 included genes are described in *Supplementary file 1*. Each well of the TaqMan Array Plate was reconstituted using a mix of Fast Master Mix and a cDNA sample (20 ng) to a set final volume (10 µl). Five plates were used for each group (CTRL, RG-7112 and o-Vanillin), and fold-change in gene expression was calculated using the $2^{-\Delta\Delta Ct}$ method [85] after normalizing to the housekeeping gene and vehicle-treated cells.

## Bioinformatics analysis

We conducted a gene expression study of a pre-specified set of apoptotic and senescence-genes of interest (Custom Taqman 96-Well Fast Plates, Thermofisher scientific) in nucleus pulposus cells. This single-gene approach offers the advantage that highly relevant genes can be identified and tested first. The candidate genes examined with this approach allowed us also to identify the genes within the selected group that together carry out the drug response in disc cells.

In order to mine the feature genes from different datasets, a Venn diagram analysis was conducted using Venny version 2.1.0 software (*Oliveros, 2007*). Differences in the expression levels of DEGs for treated and untreated groups were obtained, and the number of DEGs upregulated and downregulated were calculated. The odds ratio (OR) was calculated according to OR = (DEGs_o-

Vanillin * DEGs_RG-7112/DEGs_o-Vanillin * NonDEGs_RG-7112) / (NonDEGs_o-Vanillin * DEGs_RG-7112/nonDEGs_o-Vanillin * NonDEGs_RG-7112).

Heatmaps of gene expression pattern were constructed using unsupervised hierarchical clustering using Euclidean distance metric and complete linkage clustering method of the 43 differentially expressed genes in response to either RG-7112 or o-Vanillin ($p<0.05$) treatment of degenerate NP cells pellet. Differences in gene expression between o-Vanillin, RG-7112 and control groups were respectively compared via unpaired t-tests using the R package. Genes for which met the $p<0.05$ cut-off point were selected as DEGs, following which gene expression profiles of DEGs were visualized (heatmaps) via the 'ggplots' in R package [87] and were represented in the heatmaps as Z-scores, which is: (expression value - mean expression value across samples)/divided by the standard deviation. Colors, ranging from blue to grey then red, for each treatment represents the average fold change of each subject in that group.

Differentially expressed genes were subjected to Ingenuity Pathways Analysis (IPA) (Ingenuity Systems, Redwood City, CA) and used as a starting point for building biological networks. This analysis uses computational algorithms to identify networks consisting of focus genes (genes that were present in our list of 91 genes) and their interactions with other genes ('non-focused') in the knowledge base. Scores were calculated for each network according to the fit of the network to the set of focus genes and used to rank networks on the Ingenuity analysis. IPA uses the genes from the highest-scoring network to extract a connectivity pathway that relates candidate genes to each other based on their interactions. The involved function and disease significantly associated with these candidates' genes were shown. To generate the networks, significant pathways were filtered by p-value ($\alpha$)<0.05 and activation Z-score $> -2$ or $>2$, set as the cut-off values and representing a significant deactivation or activation, respectively.

Colors are based on log2 fold changes on these genes. To rank networks of the IPA, p-scores were calculated from p-values. For example, for n genes in the network and f of them are Focus Genes. The p-value is the probability of finding f or more Focus Genes in a set of n genes randomly selected from the Global Molecular Network calculated using Fisher's exact test. Since interesting p-values are typically quite low, it is visually easier to concentrate on the exponent and the p-score is defined as p-score = -log10(p-value). Networks with a score $\geq 2$ have at least 99% confidence that indicate a 1/100 chance that the focus genes are in a network because of random chance.

## Ex-vivo organ culture and MRI analysis

Intact lumbar spines were x-rayed, and discs were selected for the study based on the grading system described by Wilke et al. (*Page et al., 1993*). Discs with a Wilke grade of 2 were included for this study. Three IVDs from the same spine (n = 4 spines, 12 discs) were isolated and cultured as previously described (*Gawri et al., 2011*; *Krock et al., 2014*; *Rosenzweig et al., 2016*). Disc characteristics are described in *Table 3*. Isolated discs were scanned by MRI before and after treatment at day 28 as described by *Rosenzweig et al., 2018*. Images were obtained on a 7T Bruker BioSpec 70/30 USR (Bruker Biospin, Milton, ON, Canada) with the high-performance mini-imaging kit gradient upgrade AVIII electronics (Bruker) and a Bruker-issued T1ρ-RARE pulse sequence, as previously established (*Rosenzweig et al., 2018*; *Mulligan, 2015*). Briefly, 3D images were acquired and T1ρ values were quantified using the MIPAV software (NIH Center for Information Technology, Bethesda, MD, USA). T1ρ values of the same region of interest (ROI) of 'before' and 'after' treatment scans of each disc were normalized to the surrounding culture medium using editing features in MIPAV software. After 4 days of culture and one media change, single injection of the discs with vehicle, RG-7112 (5 µM/g disc) or 100 µM o-Vanillin (100 µM/g disc) in a total volume of 200 µl PBS was performed as previously described (*Krock et al., 2014*; *Rosenzweig et al., 2016*). Discs were then cultured in DMEM supplemented with 1x Glutamax, 50 µg/ml gentamicin and 1% FBS for 28 days. Media was changed and collected every 4 days. On day 28 and after MRI imaging, discs were prepared for immunohistochemistry as described below (See Intact IVD Tissue Immunohistochemistry). Conditioned media was collected at each change and frozen as individual samples at −80°C for protein analysis.

Preliminary measures to delimitate the NP region of interest (ROI) were calculated using 11-disc images from random organ donors. A contour was drawn around the disc using a Wacom Intuos Pro tablet and stylus (Wacom, Japan). This polygon was used to measure total disc voxel intensity. The NP area was created by reshaping the original disc contour to 30% of its frontal size and 40% of its

sagittal size centered around the middle of NP area, which was calculated to be 10% shift below the center of the disc (*Figure 2—figure supplement 1B-D*). The T1ρ images were manually cropped around the perimeter of the IVDs. The average of the T1ρ values was calculated within the ROIs for the vehicle and the injected discs slices per image. Heat maps representing signal intensity were created using the MIPAV software.

Age of an individual donor is specified in years. The level of the disc injected with either DMSO, O-Vanillin, and RG-7112 from the lumbar region is indicated. Discs were graded based on *Wilke et al., 2006* (*Page et al., 1993*). Disc height was determined by averaging the dorsal, ventral and midsection disc height. No differences in grade or height were found between groups prior to treatment.

## Human cytokine array of pellet and intact IVD culture media

Media from the Degenerate NP Pellet used in gene expression analysis was collected from day 4 to day 21 and pooled separately for each group (vehicle, o-Vanillin and RG-7112) and subject. Then, they were analyzed and compared to evaluate the effect of the two senolytics on SASP release. Disc media collected from day 4 (pre-treatment) and day 28 (post-treatment) were analyzed separately for each disc and each factor. The Human Cytokine Antibody Array C5 (RayBiotech, Inc) was used for semi quantitative detection of 80 proteins according to manufacturer's instructions and as previously described (*Krock et al., 2014*). Intensity units were detected by the chemiluminescence using an ImageQuant LAS4000 Image Analyzer (GE Healthcare, Baie d'Urfe, QC, Canada) and analyzed with ImageQuant TL array analysis software (GE Healthcare). The relative quantity of each factor present in each media sample was normalized to the positive and negative controls included on the array. Mean relative concentration of each factor of treated and control groups were then calculated. Data was normalized to secretion of vehicle injected discs from the same donor spine. A list of included cytokines is provided in *Supplementary file 2*.

## Luminex mulitplex assay of pellet and intact IVD culture media

Nineteen proteins were selected for analysis by Luminex multiplex assay according to manufacturer's instructions. A limitation in the number of factors we could measure was the incompatibility of some factors to be measured simultaneously as indicated by the supplier. Concentrations (pg/mL) (INF-γ, TNF-α, IL-1α, IL-1β, IL6, IL8, CCL5, CCL7, CCL11, CCL24, CCL26, CXCL1, CXCL5, CXCL9, CXCL10, CXCL11, CX3CL1, VEGF-A and Angiogenin) were measured in 40 µl media. Median fluorescence intensity (MFI) from microspheres was acquired with a BD FACSCanto II and analyzed in FlowCytomix Pro2.2.1 software (eBioscience). Concentration of each analyte was obtained by interpolating fluorescence intensity to a seven-point dilution standard curve supplied by the manufacturer.

## Intact IVD tissue immunohistochemistry

For intact IVDs, post-MRI analysis, a 2 mm wide sagittal tissue segment from the center of the IVD was fixed in periodate lysine paraformaldehyde (PLP) fixative overnight at 4°C. Samples were then washed in PBS and decalcified using Shandon TBD1 Decalcifier solution (ThermoFischer Scientific) over 72 hr at 4°C, changing solution each day. Tissue segments were washed in PBS and placed in 70% ethanol prior to paraffin embedding. Sections of 5 µm were cut and mounted on glass slides. All sections were heated on a hot plate at 55°C for 45 min and deparaffinized and rehydrated. Next, sections were stained with safranin-O/fast green (Sigma-Aldrich, Oakville, ON, Canada) and with antibodies against p16$^{Ink4a}$ and Ki-67 and counter stained using the DAB detection IHC Kit (ab64264, Abcam, Cambridge, MA) following the manufacturer's instructions. All images were acquired using a Zeiss Axioskop 40 and an AxioCam MR (Zeiss) and processed using AxioVision LE64 software (Zeiss).

## Statistical analysis

The data was analyzed using Graph Prism 8 (Graph Pad, La Jolla, CA). Analysis was performed by two-tailed Student's t test for comparison between two groups and a multiple pairwise comparison (Analysis of Variance (ANOVA) was used to evaluate the variance between multiple groups with Turkey's post hoc test. A p value < 0.05 was considered statistically significant.

## Acknowledgements

This research was funded by the Canadian Institutes of Health Research (CIHR), grant CIHR MOP-119564, a major infrastructure grant and two postdoctoral fellowships to Dr. Cherif from Arthritis Society (AS) and Réseau de Recherche en Santé Buccodentaire et Osseuse (RSBO). The authors would like to acknowledge the contribution of Kai Sheng in the experimental work and Alain Pacis from the Canadian Centre for Computational Genomics (C3G) in the figure's preparation.

## Additional information

### Funding

| Funder | Grant reference number | Author |
|---|---|---|
| Canadian Institutes of Health Research | CIHR MOP-119564 | Lisbet Haglund |
| Arthritis Society | TPF-19-0513 | Hosni Cherif |
| Réseau de Recherche en Santé Buccodentaire et Osseuse | | Hosni Cherif |

The funders had no role in study design, data collection and interpretation, or the decision to submit the work for publication.

### Author contributions

Hosni Cherif, Conceptualization, Data curation, Formal analysis, Validation, Investigation, Methodology, Writing - original draft, Writing - review and editing; Daniel G Bisson, Data curation, Formal analysis, Validation, Investigation, Visualization, Writing - review and editing; Matthew Mannarino, Validation, Investigation, Writing - review and editing; Oded Rabau, Resources, Validation, Writing - review and editing; Jean A Ouellet, Resources, Funding acquisition, Validation, Project administration, Writing - review and editing; Lisbet Haglund, Conceptualization, Supervision, Funding acquisition, Validation, Visualization, Methodology, Project administration, Writing - review and editing

### Author ORCIDs

Hosni Cherif (ID) https://orcid.org/0000-0002-0703-3898
Daniel G Bisson (ID) https://orcid.org/0000-0003-4812-059X
Lisbet Haglund (ID) https://orcid.org/0000-0002-1288-2149

### Ethics

Human subjects: All procedures are approved by and performed in accordance with the ethical review board at the RI-MUHC (IRB#sTissue Biobank2019-4896Extracellular Matrix 2020-5647) . Familial consent was obtained for each subject.

### Decision letter and Author response

Decision letter https://doi.org/10.7554/eLife.54693.sa1
Author response https://doi.org/10.7554/eLife.54693.sa2

## Additional files

### Supplementary files

• Supplementary file 1. Gene names (aliases) and primers.

• Supplementary file 2. List of cytokines and synonyms.

• Supplementary file 3. List of differentially and non-differentially expressed genes following treatment with o-Vanillin and RG-7112. (a) Eight Differentially Expressed genes significantly up- or down-regulated at p < 0.05 in RG-7112 condition. (b) Forty Differentially Expressed genes significantly up-

or downregulated at p < 0.05 in o-Vanillin condition. (c) Non-Differentially Expressed genes in RG-7112. (d) Non-Differentially Expressed genes in o-Vanillin.

• Supplementary file 4. O-Vanillin and RG-7112 IPA network scores, molecules and top diseases. (a) O-Vanillin network scores, molecules and top diseases. Physical function analysis using IPA of o-Vanillin treated NP pellets generated three networks which are ordered by a score denoting significance. The highest-scoring network, which comprises 37 proteins in our list, revealed significant changes in cell death and survival, neurological disease, organismal Injury and abnormalities. In addition, the other networks revealed changes in cancer, cellular movement, connective tissue development and function, cell cycle, organismal injury and abnormalities. (b) RG-7112 network scores, molecules and top diseases. Physical function analysis using IPA of RG-7112 treated NP pellets generated a single network which is ordered by a score denoting significance of 21. This network reveals significant changes in cell cycle, cell death and survival, connective tissue development and function.

• Transparent reporting form

## Data availability

All data generated or analysed during this study are included in the manuscript and supporting files. Source data files have been provided for all figures and figure supplements.

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
