## [Decision Letter]

**Acceptance summary:**

This paper provides ex vivo experimental evidence to demonstrate the potential of senolytic drugs as a therapeutic strategy against intervertebral disc degeneration (IDD) in humans. It is a step forward for testing senolytic drugs against aging-related disease in humans.

**Decision letter after peer review:**

Thank you for submitting your article "Senotherapeutic drugs for human intervertebral discs degeneration and low back pain" for consideration by *eLife*. Your article has been reviewed by two peer reviewers, and the evaluation has been overseen by a Reviewing Editor and Jessica Tyler as the Senior Editor. The reviewers have opted to remain anonymous.

The reviewers have discussed the reviews with one another and the Reviewing Editor has drafted this decision to help you prepare a revised submission.

Summary:

The manuscript by Haglund and colleagues seeks to test the potential of senolytic drugs as a therapeutic strategy against intervertebral disc degeneration (IDD) in humans. Using two senolytic compounds, o-Vanillin and RG-7112, the authors demonstrated their abilities to selectively kill senescent cells and to reduce the senescence-associated secretory phenotype (SASP) in a cell culture model of IDD, and discovered the gene regulatory networks that are common and distinct between the two drugs by transcriptome analysis. Furthermore, using ex vivo model of IDD, the authors showed similar senolytic effects to those found in in vitro model. The study presents a well-designed work, particularly in using the ex vivo model of IDD to demonstrate the physiological relevance and the potential of senolytics against IDD in humans, which will certainly be of general interest to the community of researchers studying the biology of aging. Both reviewers agreed that the study provides interesting insights into senescence and senolytics in IDD, and should be of broad interest. However, as detailed below, a number of concerns were also raised, relating in large part to insufficient clarity in the current version with respect to the authors' methods and the limitations of their approach and data presentation/interpretation, and the rationale behind the choice of drugs.

Essential revisions:

1) The rationale on the choice of two drugs should be clarified, as it seems that they are "arbitrarily" included together in the same manuscript. In particular, since the authors have already demonstrated the senolytic effects of o-Vanillin in in vitro IVD model, what was the rational for showing the effects of yet another drug, RG-7112, in a descriptive way rather than delving deeper into the mechanisms of o-Vanillin? Is one better than the other? Would they be expected to target fundamentally unique and complementary pathways? In light of the recent studies showing that the combination of senolytic drugs, e.g. dasatinib and quercetin (D+Q), selectively targets a broader range of senescent cell types than alone (Xu et al., Nature Medicine, 2019), why not testing the potential of the combo treatment?

2) The relationship and rationale of comparing metabolic activities measured at day 4 of treatment vs. assessments of senescence measured at day 21 of treatment need to be clarified.

3) The descriptions on transcriptome data analysis are inadequate and need improvement as follows:

a) It needs more information as to how the transcriptome analysis was performed in the Materials and methods section. It is not even clear if it was genome-wide or candidate approach. Description of samples, materials, and basic workflows are needed, and information on data availability (GEO and accession number) should be provided.

b) It needs more reasoning and better interpretation to enhance readability to non-specialists in the field of cellular networks and bioinformatics.

c) While information on IVD samples are tabled, it would be important to provide a general logic and information on grouping of the different tissue/cell samples for each of the analysis. For examples, in each of the comparative analyses, were the same cells used in the control and tested experiments? If they were different, how would this affect the interpretation and statistical analyses? In Figure 1—figure supplement 1, it is difficult to locate data generated from different sources and compared with n=6, 3 and 8, and the degenerative score of the cells and tissues etc.

d) The presentation of the genes and gene numbers from each of the treatments with RG-7112 or o-Vanillin were complex and the information provided is not easy to follow. The authors concluded that distinct senescence pathways are affected based on 8 genes affected by RG-7112 treatment and 40 genes by o-Vanillin, but only 2 genes common to both. They need to provide more information to better understand the results. For example, in Figure 2, a volcano plot and a venn diagram will be useful with statistical test to assess the overlaps. In Figure 2A, how are the genes in heat map ordered vertically; from hierarchical clustering or just shown as a dendrogram? Further, how was the clustering performed; using Spearman or Euclidean, and the linkage (average or complete)? The information provided in subsection “Bioinformatics Analysis” is too vague.

e) It was also unclear which gene sets were used for the IPA, the entire RNA-seq data or a selection that also included the differentially expressed genes. From the data and networks provided, the authors need to state clearly what the new insights, as they appear to be the expected pathways, were. The information provided in the Results section on the IPA pathways are superficial. IPA is very generic, it does not consider species, tissue, development stage, disease/control etc.; thus cautions need to be taken in not over-interpreting the IPA. Importantly, statistics is needed to support the conclusions. For example, what are the P-values of the networks in Figure 2 (B)-(E)? Are they the most significant ones? Were there others that you didn't show? The authors need to clarify these points in the Materials and methods.

f) Finally, it should be avoided to make conclusive statements when there is only a trend without statistical significance.

4) The descriptions on the SASP analysis need clarifications as follows:

a) Logic of the data presentation in Figure 3 is confusing and inconsistent. The authors should provide a clear picture as to the number of factors tested, affected, and chosen to be further followed-up and with proper statistical analysis. Please provide a scatter-plot of the SASPs' average changes per drug in Figure 3A, in order to show potential of co-linearity in the responses. Explain why the Luminex assay was used to reassess the 19 factors: Is this more accurate and quantitative, and how are the two assay correlated?

b) The heat map of the 19 gene panel in Figure 3B suggests similar and different patterns of change between the two treatments. Provide information on age, sex, degenerative score, and other IVD characteristics on the two individuals in the RG-7112 treatment that seem to be different. Are the same set of cells used for the respective control and treatments? Such information seem to be missing and nor discussed in the manuscript.

c) The gene information obtained seem not have contributed to additional analyses, as the authors then moved specifically to an analysis of SASPs which were not implicated in the network data. Since they indicate "To verify the effect of RG-7112 and o-Vanillin on SASP factor release…", perhaps the authors could make some links here to SASP to present a better flow and logic. Again, please state whether the same cells were used in the control, RG7112 and o-Vanillin assessments. This would be important in the interpretation of the gene differences.

5) For the T1p-weighted MRI and disc matrix data, more description and clarity on the organ culture and data analysis need to be provided. For example, are the MRI and immunostaining data from a single injection? Further, were the medium collected from these discs for the measurement of SASP shown in Figure 5? Statistical analysis should be performed to support the conclusions made.

[Editors' note: further revisions were suggested prior to acceptance, as described below.]

Thank you for resubmitting your work entitled "Senotherapeutic drugs for human intervertebral disc degeneration and low back pain" for further consideration by *eLife*. Your revised article has been evaluated by Jessica Tyler (Senior Editor) and a Reviewing Editor.

The authors have made concerted effects to constructively address the concerns and criticisms raised by two reviewers who agree that the revised manuscript has improved. However, reviewer #2 still has questions and concerns which are mostly pertinent to gene expression analysis. Before publication, these should be addressed to further improve the clarity and transparency of the manuscript. I would like to encourage the authors to revise the manuscript to address the specific suggestions and questions raised by reviewer 2. My recommendation is as follows:

1) Clarify gene expression variability vs genetic variation: In the methodology under the "bioinformatics analysis", the author indicated they "conducted a gene association study that focus at the genetic variation associated with degenerate nucleus pulposus cell". This is not a genetic study but a study in gene expression?

2) Describe selection criteria and provide the full list of 91 genes in supplement: It is also stated that it is "a pre-specified set of apoptotic and senescence-genes of interest". How were the 91 genes selected and a need to show the full list? As this are preselected, there would be a bias on the potential pathway networks, and thus the differentially expressed genes are confined within this gene set and one would expect specific pathways to be within this set. Therefore, it would be appropriate to disclose the existing pathways within this set of 91 genes, and estimate the selectivity of this pathway in the DEGs for RG-7112 or O-Vanillin.

3) Provide a line or two justification, e.g transcriptome analysis provides candidate gene sets and shorten the discussion around the DEGs: I do not think the transcriptome data added value but rather confusion as the gene set were pre-selected for apoptosis and senescence pathways, and the need to study SASP does not seem to be directly linked to this dataset. Therefore, I would be rather careful in placing a lot of weight on this data, as much of the discussion evolved around the DEGs.

4) Rephrase the sentences not to mislead: I still have concern that the authors continue to stretch the boundary of suggestive results that have not reach statistical significance. Suggesting a trend in the absence of statistical support is not scientific. Thus statements such as "approaching significance" and "slightly non-significant increase" should be avoided, as the data presented indicated there is not difference. Once can never predict how the values will changed with more data point.

5) Indicate that they were selected as candidates from the transcriptome analysis: In the main text, it is described as "RNA-Seq analysis". This is not correct, as the authors should clearly indicate they are not testing RNA-Seq genome-wide, but only 91 selected genes. Very confusing.

6) Describe in the text the nature of the 4 overlapped genes and their directional impact: In Figure 2A, is the venn diagram included both up and down-regulated DEGs? May consider providing separate Venn diagrams, for up- and down-regulated genes. Also, need to provide an odds-ratio for the overlaps, and a p-value for the Chi-square test of co-occurrence. A crude estimate is that by chance alone, if you randomly sample 8 genes from 91, and another 40 from 91, you will get 8*40/91=3.51, and in reality you detect 4 genes in overlap, which is not much more than by chance. Further, should indicate the identity of these 4 overlapping genes.

7) Clarify if FDR or p values were used for cut-off for DEGs: It was indicated a p<0.05 was used as cutoff for DEGs, "Of the 91 genes tested, 44 were differentially expressed with a P<0.05 in one or both treatments (Figure 2B)". But in Figure 2C, the author indicated using "logFDR". Therefore, which measure was used for the cutoff?

8) In the figure legend, Y axis is indicated as log10 of the p-values while in the Figure 2C, log (FDR). Clarify which one: Remake the Figure 2C so that two drug effects can be easily differentiated: For the vertical axis in Figure 2C, is it log(FDR) or log10FDR ? as typically, "log10FDR" is used. In Figure 2C, genes can be better illustrated with different colours for gene with logFC >0 and those <0, and different shape for the drugs, or separate volcano plots. For insignificant genes, use grey. Further, this figure is problematic to me; as it does not seem to reflect a typical volcano plot. If FDR used, then many of the non-significant p-values would be 1.0, and the Log of 1 is zero. So, many more of the values should be around zero. Some explanation needed.

9) It is OK to leave as they are: For Figure 2(D) to (G), for each of these networks, how unusual is it to observe this? Given the bias, you are bound to have some networks, even by randomly selecting genes into IPA. Can the authors provide the p values or FDR to address this, and label them on the figures directly?

Reviewer #1:

I have no further comments. The authors admirably addressed my concerns and should be commended for a careful and thorough revision.

Reviewer #2:

This revised manuscript has improved in logic and additional information provided have addressed most of my concerns. However, I am still uncertain on the bioinformatics approach for the transcription data that remained unclear to me, and the potential bias in the approach and interpretation.

In the methodology under the "bioinformatics analysis", the author indicated they "conducted a gene association study that focus at the genetic variation associated with degenerate nucleus pulposus cell". This is not a genetic study but a study in gene expression?

It is also stated that it is "a pre-specified set of apoptotic and senescence-genes of interest". How were the 91 genes selected and a need to show the full list? As this are preselected, there would be a bias on the potential pathway networks, and thus the differentially expressed genes are confined within this gene set and one would expect specific pathways to be within this set. Therefore, it would be appropriate to disclose the existing pathways within this set of 91 genes, and estimate the selectivity of this pathway in the DEGs for RG-7112 or O-Vanillin.

I do not think the transcriptome data added value but rather confusion as the gene set were pre-selected for apoptosis and senescence pathways, and the need to study SASP does not seem to be directly linked to this dataset. Therefore, I would be rather careful in placing a lot of weight on this data, as much of the discussion evolved around the DEGs.

I still have concern that the authors continue to stretch the boundary of suggestive results that have not reach statistical significance. Suggesting a trend in the absence of statistical support is not scientific. Thus statements such as "approaching significance" and "slightly non-significant increase" should be avoided, as the data presented indicated there is not difference. Once can never predict how the values will changed with more data point.

In the main text, it is described as "RNA-Seq analysis". This is not correct, as the authors should clearly indicate they are not testing RNA-Seq genome-wide, but only 91 selected genes. Very confusing.

In Figure 2A, is the venn diagram included both up and down-regulated DEGs? May consider providing separate Venn diagrams, for up- and down-regulated genes. Also, need to provide an odds-ratio for the overlaps, and a p-value for the Chi-square test of co-occurrence. A crude estimate is that by chance alone, if you randomly sample 8 genes from 91, and another 40 from 91, you will get 8*40/91=3.51, and in reality you detect 4 genes in overlap, which is not much more than by chance. Further, should indicate the identity of these 4 overlapping genes.

It was indicated a p<0.05 was used as cutoff for DEGs, "Of the 91 genes tested, 44 were differentially expressed with a P<0.05 in one or both treatments (Figure 2B)". But in Figure 2C, the author indicated using "logFDR". Therefore, which measure was used for the cutoff?

For the vertical axis in Figure 2C, is it log(FDR) or log10FDR ? as typically, "log10FDR" is used. In Figure 2C, genes can be better illustrated with different colours for gene with logFC >0 and those <0, and different shape for the drugs, or separate volcano plots. For insignificant genes, use grey. Further, this figure is problematic to me; as it does not seem to reflect a typical volcano plot. If FDR used, then many of the non-significant p-values would be 1.0, and the Log of 1 is zero. So, many more of the values should be around zero. Some explanation needed.

For Figure 2(D) to (G), for each of these networks, how unusual is it to observe this? Given the bias, you are bound to have some networks, even by randomly selecting genes into IPA. Can the authors provide the p values or FDR to address this, and label them on the figures directly?

---

## [Author Response]

Essential revisions:1) The rationale on the choice of two drugs should be clarified, as it seems that they are "arbitrarily" included together in the same manuscript. In particular, since the authors have already demonstrated the senolytic effects of o-Vanillin in in vitro IVD model, what was the rational for showing the effects of yet another drug, RG-7112, in a descriptive way rather than delving deeper into the mechanisms of o-Vanillin?

We appreciate the reviewer’s insight. It is proposed that, the inflammatory environment triggered by senescent cells prevents adjacent cells from maintaining tissue homeostasis (1-3) and it is proposed to induce senescence in a paracrine manner thus exacerbating tissue deterioration (4). The rational for comparing the two drugs was to evaluate if natural senolytics with anti-oxidant or anti-inflammatory properties (like o-Vanillin) in addition to their senolytic effect further reduces inflammatory factors released by non-senescent cells thus enhancing the therapeutic effect (5-8). RG-7112, a pure senolytic without documented anti-inflammatory effects, was used for comparison. Another reason to test the two different drugs was to evaluate bioavailability to cells in their native environment. It was not known if either of the drugs could reach and kill senescent cells embedded in their native environment. A reason to select RG7112 instead of other available senolytic agents was that it is an analogue of UBX0101, a nutilin-3a inhibitor that also targets the p53/MDM2 interaction (9-10). UBX0101, is currently in clinical trials to treat knee osteoarthritis, it promotes clearance of senescent cells in cartilage from patients with knee osteoarthritis (9, 11). Articular cartilage and IVD tissue share many characteristics both at the environmental, cellular and molecular level and we hypothesized that a drug effective in articular cartilage would have a good chance of also working in the IVD. RG-7112 is an FDA approved drug (12) and was the first MDM2 inhibitor to be advanced into Phase I human clinical trials (NCT01164033, NCT01143740, NCT00623870, NCT00559533), to treat a wide range of cancers (13-16). Altogether, the similarities to UBX0101, the safety and promising clinical properties of RG-7112 as well as repurposing this approved drug may offer an efficient drug development pathway for treatments of intervertebral disc degeneration and low back pain with few or no therapeutic options. We clarified the rationale for choosing the two drugs in the Introduction.

Is one better than the other? Would they be expected to target fundamentally unique and complementary pathways?

The pathway affected by RG7112 was known while the mode of action was unknown for o-Vanillin. The two major pathways mediating cell-cycle arrest during disc degeneration are p53-p21-Rb and p16-Rb (17). Although both drugs activate cyclin-dependent-kinases (CDKs: cdk4 and cdk6) directing senescent cells to apoptosis and nonsenescent cells to proliferation, they act on two different upstream pathways. RG-7112 act on the MDM2-p53 interaction to reduce p21 accumulation leading to the suppression of its inhibitory effect of CDKs. This was confirmed in the pathway analysis. The data presented here is showing that o-Vanillin act by reducing p16 and suppressing the inhibitory effect on CDKs. Our findings suggest that the compounds result in a similar end result that can be explained by the activation of the same downstream effectors: CDKs-E2F-RB (please see Author response image 1). At this level of the study, our first and most important goal was to determine that the drugs could reach the target cells and remove senescent cells and associated inflammatory factors in native intact human IVD tissue.

**Author response image 1. sa2fig1:** 

In light of the recent studies showing that the combination of senolytic drugs, e.g. dasatinib and quercetin (D+Q), selectively targets a broader range of senescent cell types than alone (Xu et al., Nature Medicine, 2019), why not testing the potential of the combo treatment?

It is possible that the combination would have a stronger effect. However, we found that the drugs removed both senescent NP and AF cells. Xu et al. used Dasatanib and Quercetin since the senescent cell associated pathways (SCAPs) of senescent preadipocytes differed considerably from those of senescent endothelial cells and that one drug could not remove both types (18). Similar navitoclax, a BCL‐2 prosurvival pathway inhibitor, lacks senolytic effect in adipose tissue, but displays a strong effect on senescent endothelial cells and fibroblasts (19-20). Combining drugs allowed targeting of heterogenous senescent cell populations in conditions where they are spread across multiple organs (6). For example, ongoing clinical trials using systemic administration of D+Q include diseases with senescent cells in multiple tissues such as pulmonary fibrosis, chronic kidney disease and hematopoietic stem cells. For local osteoarthritis treatment, a single drug, UBX0101, was used to target senescent chondrocytes (8). In our study we aimed to target localized natural occurring senescent disc cells using local administration of o-Vanillin or RG-7112. We chose to initially test them separately as the supply of live intact human IVDs are very limited. Further, single treatment may reduce side effects like those frequently reported for D+Q therapy, including respiratory symptoms and gastrointestinal discomfort or heartburn (21). We agree that it will be important to evaluate the combination especially in vivo. Preclinical animal experiments will be performed to optimize frequency of treatment as well as to verify safety and tolerability, and determine specific, senolytic effects of single or combination treatments for intervertebral disc degeneration and low back pain.

2) The relationship and rationale of comparing metabolic activities measured at day 4 of treatment vs. assessments of senescence measured at day 21 of treatment need to be clarified.

We apologize for the lack of clarity regarding the relationship between metabolic activity and the assessments of senescence. No measure was performed at 4 days, both metabolic activity and assessments of senescence were performed at 21 days for disc cell cultured in pellet. Metabolic activity, measured in pellet cultures, was used to determine a safe dose without cytotoxicity to non-senescent cells (Figure 1—figure supplement 1A). We treated the cells for a four-day period. Cytotoxicity and senolytic effect of the treatment, was then evaluated at day 21 to allow for the drugs to take effect and also to verify a sustained effect Figure 1(A-D). In Monolayer culture, the rationale behind the comparison of the effect of the treatment in the cells from degenerate and non-degenerate discs is to confirm the senolytic activity (Figure 1—figure supplement 1E) and the safety of RG-7112 (Figure 1—figure supplement 1F). We modified the text accordingly in the Results and the Materials and methods sections to clarify.

3) The descriptions on transcriptome data analysis are inadequate and need improvement as follows:a) It needs more information as to how the transcriptome analysis was performed in the Materials and methods section. It is not even clear if it was genome-wide or candidate approach. Description of samples, materials, and basic workflows are needed, and information on data availability (GEO and accession number) should be provided.

We thank the reviewers for the constructive comment. We have now further described the samples, materials and workflow in the Materials and methods section of the revised manuscript and in the legend of Figure 2. This small dataset of a candidate approach might not need GEO submission as all data related to Figure 2 is now available in the attached source data files of Figure 2: Figure 2—source-data 1, Figure 2—figure supplement 1—source-data 1 and the Supplementary File 3 (a-d). In brief, we conducted a candidate gene association study to evaluate the variation in gene expression profiles associated with treated and non-treated cells within a pre-selected set of apoptotic and senescence-genes of interest. This approach offers the advantage that highly relevant genes can be identified and tested first. The candidate genes examined in this pathway approach allowed us also to identify the genes and aggregate of genes that together carry out the drug response of disc cells to the senolytic treatment.

b) It needs more reasoning and better interpretation to enhance readability to non-specialists in the field of cellular networks and bioinformatics.

We agree with the reviewers comment and have modified the text in the Results section accordingly.

c) While information on IVD samples are tabled, it would be important to provide a general logic and information on grouping of the different tissue/cell samples for each of the analysis.

We acknowledge this concern. In Table 2, we linked each result of the assays, cultures and cell type used as well as disc level and the donor from which the samples were obtained. To more precisely clarify the link between the tissue and cells used with the results, we indicated in Table 2 the degeneration score (according to Thompson grading scale) and state (degenerate and non-degenerate) of each of the intact discs. We evaluated cells from degenerate discs as well as intact degenerate disc as this would correspond with the tissue targeted for treatment. In addition, it would be difficult to evaluate an effect of removing senescent cells in non-degenerate discs as they have very few senescent cells (5). The effect of the two senolytics on cells from non-degenerate discs was done to confirms their safety for healthy cells (Figure 1—figure supplement 1F). Also, we clarified the logic behind grouping different samples in the figure’s legends and our revised text and we added the donor ID number and gender for each of the heatmaps in Figure 2B, Figure 3B, Figure 5C and Figure 2—figure supplement 1-A.

For examples, in each of the comparative analyses, were the same cells used in the control and tested experiments? If they were different, how would this affect the interpretation and statistical analyses?

In the comparative analyses, we always used cells from the same donor for control and experimental conditions. the number (n) in each experiment represent the number of donors used to replicate the experiment with the same conditions (biological replicates).

In Figure 1—figure supplement 1, it is difficult to locate data generated from different sources and compared with n=6, 3 and 8, and the degenerative score of the cells and tissues etc.

The data generated in Figure 1—figure supplement 1 is obtained from independent experiments, using complementary assays, to demonstrate the senolytic activity of RG-7112. Thus, they cannot be directly compared. Figure 1—figure supplement 1A is verifying that the RG-7112 concentration selected from the literature (22-23) has no cytotoxic effect on human IVD cells. Metabolic activity is compared between RG7112 treated and untreated cells from 6 donors (Table 2). Figure 1—figure supplement 1B and 1C demonstrate the apoptotic effect RG-7112 in NP cells from 3 donors. Figure 1—figure supplement 1D illustrates the senolytic activity of RG-7112 by immunostaining, where apoptotic caspase 3 positive cells are also p16*^ink4a^* positive while proliferating ki-67 positive cells are not p16*^ink4a^* positive (NP cells from 3 different donors). Figure 1—figure supplement 1E confirm the apoptotic effect of RG-7112 observed in Figure 1—figure supplement 1B and 1C by an increase of caspase3/7 activity in degenerate NP cells, where there are more senescent cells compared to non-degenerate cells. Figure 1—figure supplement 1F validate the safety of RG-7112 in both degenerate and non-degenerate cells by showing similar metabolic activity. In Figure 1—figure supplement 1E-1F, NP cells used were from 4 degenerate and 4 nondegenerate discs for a total number n = 8. In Figure 1—figure supplement 1G we used pellet culture media of NP and AF cells from 6 donors to demonstrate the beneficial effect of the treatment with RG-7112 on proteoglycan synthesis (sGAG release). The degenerative state for each disc used in each of the assays presented in Figure 1—figure supplement 1 are included in Table 2 in the revised manuscript. We modified the text accordingly in the legend of Figure 1—figure supplement 1, the Results and the Materials and methods sections in the revised version of the manuscript.

d) The presentation of the genes and gene numbers from each of the treatments with RG-7112 or o-Vanillin were complex and the information provided is not easy to follow. The authors concluded that distinct senescence pathways are affected based on 8 genes affected by RG-7112 treatment and 40 genes by o-Vanillin, but only 2 genes common to both. They need to provide more information to better understand the results. For example, in Figure 2, a volcano plot and a venn diagram will be useful with statistical test to assess the overlaps.

We fully agree with this request and accordingly we included new figures and tables to clarify the information provided. Please see Figure 2A, Figure 2C and the Supplementary File 3 (a-d). Better interpretation of the results is now included in the new version of the manuscript (please see response in 3. 2).

In Figure 2A, how are the genes in heat map ordered vertically; from hierarchical clustering or just shown as a dendrogram? Further, how was the clustering performed; using Spearman or Euclidean, and the linkage (average or complete)? The information provided in subsection “Bioinformatics Analysis” is too vague.

In Figure 2B, (Figure 2A in the previous version), the significant differentially expressed genes in the heat map were ordered vertically from hierarchical clustering using Euclidean distance metric and complete linkage clustering method. The text of the Materials and methods section was modified accordingly in the revised version of the manuscript.

e) It was also unclear which gene sets were used for the IPA, the entire RNA-seq data or a selection that also included the differentially expressed genes. From the data and networks provided, the authors need to state clearly what the new insights, as they appear to be the expected pathways, were. The information provided in the Results section on the IPA pathways are superficial. IPA is very generic, it does not consider species, tissue, development stage, disease/control etc.; thus cautions need to be taken in not over-interpreting the IPA. Importantly, statistics is needed to support the conclusions. For example, what are the P-values of the networks in Figure 2 (B)-(E)? Are they the most significant ones? Were there others that you didn't show? The authors need to clarify these points in the Materials and methods.

We fully accept this as a valid comment. in Figure 2 (D)-(G) (B-E in the old version), only the differentially expressed genes (P < 0.05) were included in the network analysis. We clarified the experimental procedures of signaling pathway analysis using IPA in the revised version of the manuscript. We also included new supplemental tables in the Supplementary File 3 (a-d) for the average fold change of all the genes (differentially and non-differentially expressed) in the o-Vanillin and RG-7112 treated NP pellets as well as their respective p values and tables in the Supplementary File 4 (a-b) for the significance scores, focus molecules used to reorder the networks and determine the possible physical function of each compound. In accordance with the reviewer’s suggestion, we have modified the IPA results, legends and discussion in the new version of the manuscript to highlight the new insights and better interpret our findings.

f) Finally, it should be avoided to make conclusive statements when there is only a trend without statistical significance.

As suggested, we modified the conclusive statements where statistical significance was not reached.

4) The descriptions on the SASP analysis need clarifications as follows:a) Logic of the data presentation in Figure 3 is confusing and inconsistent. The authors should provide a clear picture as to the number of factors tested, affected, and chosen to be further followed-up and with proper statistical analysis. Please provide a scatter-plot of the SASPs' average changes per drug in Figure 3A, in order to show potential of co-linearity in the responses.

We agree the reviewer’s comments. Therefore, we added in Figure 3—figure supplement 1A, in our revised manuscript, a descriptive schematic to summarize the number of factors tested, factors detectable by the cytokine array and factors selected for Luminex assay. We modified the text accordingly in the new version of the manuscript to better describe the analysis and discuss the results. The complete list of the factors analyzed by cytokine array and Luminex assay in Figure 3, Figure 5, Figure 3—figure supplement 1 and Figure 4—figure supplement 1are included in the corresponding source data files uploaded with the manuscript. As suggested, we revised Figure 3—figure supplement 1A and classified the SASP factors in 4 groups: cytokines, chemokines (CC and CXC series), growth and neurotrophic factors and we showed the most downregulated factors in each class. Also, to better visualize the average effect of the two drugs on SASPs release, a scatter plot representing the average change for 50 factors is included in the new version of the manuscript in Figure 3A (e) and subsection “RG-7112 and o-Vanillin reduced inflammatory factors”. Statistical analysis was performed for both cytokine array and Luminex analysis in pellets and discs cultures. Significances, is highlighted in the figures and their respective legends when applicable.

Explain why the Luminex assay was used to reassess the 19 factors: Is this more accurate and quantitative, and how are the two assay correlated?

Both used the same media from the same donors. Cytokine array was used to semi-quantitative screen 80 factors. Most factors were modulated by the two senolytics (in both pellets and intact disc). Luminex assay is more sensitive and it allowed us to measure the concentrations of selected factors in all conditions. The choice of the 19 factors were based on the cytokine array as follows: only factors detected in all conditions and in at least four donors were chosen. These factors were selected to represent a variety of SASP factors and to include the most variably expressed cytokines, chemokines and growth factors. A limitation in the number of factors we could measure was the incompatibility of some factors to be measured simultaneously (as indicated by the supplier). The Luminex analysis reproduced the overall changes observed by the cytokine array. However, more significant differences were measured in treated vs. untreated groups. Also, Luminex results allowed us to better visualize and correlate the observed similarities and differences in amplitude and significance between the two compounds for each factor analyzed.

b) The heat map of the 19 gene panel in Figure 3B suggests similar and different patterns of change between the two treatments. Provide information on age, sex, degenerative score, and other IVD characteristics on the two individuals in the RG-7112 treatment that seem to be different. Are the same set of cells used for the respective control and treatments? Such information seem to be missing and nor discussed in the manuscript.

We agree with the reviewer comment on the variability pattern between the donors. This is expected due to the heterogenic nature of human patients including, age, sex, degeneration level, diseases, physical activity, cause of death…. However, it is important to highlight the similar average effect (downregulation of SASP factors) observed in Figure 3C and Figure 3—figure supplement 1C following RG-7112 and o-Vanillin treatment. Cells from the same donor were used for control and treatment, as now indicated by an ID number and the gender of each subject. Demographic and disc characteristics information for each donor used is provided in Table 1 and 2, and in the figure’s legends and the text of the revised version of the manuscript.

c) The gene information obtained seem not have contributed to additional analyses, as the authors then moved specifically to an analysis of SASPs which were not implicated in the network data. Since they indicate "To verify the effect of RG-7112 and o-Vanillin on SASP factor release…", perhaps the authors could make some links here to SASP to present a better flow and logic. Again, please state whether the same cells were used in the control, RG7112 and o-Vanillin assessments. This would be important in the interpretation of the gene differences.

The gene analysis was performed to verify the senolytic activity of the two compounds and the prediction of the potential pathway(s) activated. All the focus genes selected were senescence and cell cycle genes which explain the absence of SASP genes in the network data. However, the cytokine array analysis was performed to evaluate and measure, at the protein level, the effect the two compounds have on SASP factor release. In accordance with the reviewer’s request, we modified the Result and Materials and method sections in the revised version of the manuscript to better highlight the complementarity nature of the analysis. The media from the pellets used for cytokine array and Luminex assay in Figure 3 and Figure 3—figure supplement 1 were the same pellets used for gene expression analysis as mentioned in the legend of Figure 3.

5) For the T1p-weighted MRI and disc matrix data, more description and clarity on the organ culture and data analysis need to be provided. For example, are the MRI and immunostaining data from a single injection? Further, were the medium collected from these discs for the measurement of SASP shown in Figure 5? Statistical analysis should be performed to support the conclusions made.

We acknowledge these valid suggestions. Accordingly, we added Figure 4A to better describe the organ culture and data analysis. All the ex vivo analysis (in Figure 4, Figure 5 and Figure 4—figure supplement 1) was performed on the discs, and their respective media, following a single injection of the senolytic drug or vehicle at day 4. The discs used for the 3 conditions (CTRL, o-Vanillin and RG-7112) were from the same donor and the experiment was performed on IVDs from four different donors (Table 3). We modified the ex vivo organ culture and MRI analysis section in the new version of the manuscript for more clarity. Statistical tests (two tailed student t-test to compare pre and post treatment groups and ANOVA Kruskal-Wallis nonparametric test with Turkey’s post hoc test was performed for multiple pairwise comparison between treated and untreated groups. This information is highlighted when applicable in the legends of the figures and in the Materials and methods.

[Editors' note: further revisions were suggested prior to acceptance, as described below.]

The authors have made concerted effects to constructively address the concerns and criticisms raised by two reviewers who agree that the revised manuscript has improved. However, reviewer #2 still has questions and concerns which are mostly pertinent to gene expression analysis. Before publication, these should be addressed to further improve the clarity and transparency of the manuscript. I would like to encourage the authors to revise the manuscript to address the specific suggestions and questions raised by reviewer 2. My recommendation is as follows:1) Clarify gene expression variability vs genetic variation: In the methodology under the "bioinformatics analysis", the author indicated they "conducted a gene association study that focus at the genetic variation associated with degenerate nucleus pulposus cell". This is not a genetic study but a study in gene expression?

Indeed, it is a gene expression study. We modified the text of the revised version of the manuscript.

2) Describe selection criteria and provide the full list of 91 genes in supplement: It is also stated that it is "a pre-specified set of apoptotic and senescence-genes of interest". How were the 91 genes selected and a need to show the full list? As this are preselected, there would be a bias on the potential pathway networks, and thus the differentially expressed genes are confined within this gene set and one would expect specific pathways to be within this set. Therefore, it would be appropriate to disclose the existing pathways within this set of 91 genes, and estimate the selectivity of this pathway in the DEGs for RG-7112 or O-Vanillin.

A complete list of the genes is found in Supplementary File 1. The genes were provided in a configurable human TaqMan array for human cellular senescence, fast 96-well plate from Thermofisher scientific (Array ID: RPU62T7, Catalog number: 4413255). The plate selected was customized to also include recently described genes implicated in senescent cells anti- apoptotic pathways (SCAPs) (24). The assay includes 82 pre-selected genes involved in cellular senescence and 2 housekeeping genes (GAPDH and 18s). We selected an additional 12 SCAPs genes for a total of 96 gene. Out of the 96 genes analyzed, 91 were detectable and are represented in the results of Figure 2. The Assay ID is provided in the key resources table. We included this clarification in the Materials and methods section of the revised manuscript.

The Network analysis feature in Ingenuity pathway analysis (IPA) (Ingenuity Systems, Redwood City, CA) suite is a broad analysis of interactions among identified proteins mined from various literature sources: The Ingenuity pathway knowledge base (IPKB). The program does not generate potential networks as it takes into account any possible combination of up or down regulated gene combinations. It is therefore not possible to disclose potentially existing pathways within the set. IPA is used to further analyze functional associations and networks of interconnected proteins. We agree that the discovered pathways are restricted to the set of 91 genes analyzed and they reflect the specific pattern of DEGs for each compound. Using IPA, we identified genes that fell within IPA-generated pathways at a higher frequency than those expected to occur for a randomly selected set of genes. This analysis identified three interacting networks for o-Vanillin and one for RG-7112 within the selected genes list. The number of networks obtained for each compound is correlated with their respective number of DEGs (8 DEGs were involved in one network for RG-7112 (Figure 2D) while 40 DEGs for o-Vanillin were involved in three networks (Figure 2 (E-G)). Interaction networks were limited to 35 molecules per network and 25 networks per analysis and excluded endogenous chemicals. We focused on algorithmically derived interaction networks, which are assigned a score on the basis of their relevance to the genes in the input dataset, the number of focus genes (i.e. dysregulated genes in our data that are in that network), and their connectivity (25). The score is calculated as –log_10_(p value), where p is generated using a Fisher’s exact test (26). Networks with a score ≥2 have at least 99% confidence that it is not generated by chance (27). Studies have found scores greater than 3 to be significant, with a score of 3 indicating a 1/1000 chance that the focus genes are in a network because of random chance (28-30). Other studies have opted to utilize more stringent criteria and higher scores to ensure that their discovered networks are highly significant (31-32); the network score for RG-7112 was 21 and for the three o-Vanillin networks were 37, 24 and 13. The score is not an indication of the quality or biological relevance of the network; it simply calculates the approximate "fit" between each network and our Network Eligible Molecules. Moreover, the single network pathway affected following the treatment with RG-7112 in disc cells validate the expected MDM2-p53 pathway previously reported in other cell types. The complete list of networks, their scores, the focus molecules and associated diseases and functions are provided in the Supplementary File 4. We modified the text in the revised version of the manuscript to disclose the obtained pathways within the selected set of 91 genes and estimate their selectivity.

3) Provide a line or two justification, e.g transcriptome analysis provides candidate gene sets and shorten the discussion around the DEGs: I do not think the transcriptome data added value but rather confusion as the gene set were pre-selected for apoptosis and senescence pathways, and the need to study SASP does not seem to be directly linked to this dataset. Therefore, I would be rather careful in placing a lot of weight on this data, as much of the discussion evolved around the DEGs.

We agree, gene expression analysis of known senescence associated genes was performed to determine how the compounds carry out their senolytic activity. We reduced the discussion around the DEGs in the revised version.

4) Rephrase the sentences not to mislead: I still have concern that the authors continue to stretch the boundary of suggestive results that have not reach statistical significance. Suggesting a trend in the absence of statistical support is not scientific. Thus statements such as "approaching significance" and "slightly non-significant increase" should be avoided, as the data presented indicated there is not difference. Once can never predict how the values will changed with more data point

The higher variability between individual human specimens prompt us to report any trend in the results. However, we agree with the reviewer’s recommendation and modified the text of the revised manuscript to exclude any statements such as approaching significance /slightly non-significant for better scientific accuracy.

5) Indicate that they were selected as candidates from the transcriptome analysis: In the main text, it is described as "RNA-Seq analysis". This is not correct, as the authors should clearly indicate they are not testing RNA-Seq genome-wide, but only 91 selected genes. Very confusing.

We thank the reviewers for the comment. We have now corrected in the revised manuscript text.

6) Describe in the text the nature of the 4 overlapped genes and their directional impact: In Figure 2A, is the venn diagram included both up and down-regulated DEGs? May consider providing separate Venn diagrams, for up- and down-regulated genes. Also, need to provide an odds-ratio for the overlaps, and a p-value for the Chi-square test of co-occurrence. A crude estimate is that by chance alone, if you randomly sample 8 genes from 91, and another 40 from 91, you will get 8*40/91=3.51, and in reality you detect 4 genes in overlap, which is not much more than by chance. Further, should indicate the identity of these 4 overlapping genes.

The Venn diagram in Figure 2A of the previous version include both up- and down-regulated DEGs and the 4 overlapping genes are the 4 significantly DEGs: MAPK14, CDC25c, CCNB1 and CDKN2D (p19*^ARF^*) as mentioned in the Results. In the revised version of the manuscript, two separate Venn diagrams for up and down regulated DEGs is provided in Figure 2A viewer. The total number of overlapping genes is now three since CCNB1 is upregulated in RG-7112 and downregulated in o-Vanillin. In upregulated DEGs, the three overlapping genes between o-Vanillin and RG-7112 are MAPK14, CDC25c and CDKN2D (p19*^ARF^*). No down-regulated DEGs was shared between RG-7112 and o-Vanillin. Please see the modified legend and Results section. The detailed list of up- or down-regulated DEGs and their respective p values are provided in the Supplementary file 3. Odds ratios (33) and p values were calculated using Fisher exact test.

7) Clarify if FDR or p values were used for cut-off for DEGs: It was indicated a p<0.05 was used as cutoff for DEGs, "Of the 91 genes tested, 44 were differentially expressed with a P<0.05 in one or both treatments (Figure 2B)". But in Figure 2C, the author indicated using "logFDR". Therefore, which measure was used for the cutoff?

We acknowledge this concern. p<0.05 was used as a cut-off for DEGs in both Figure 2B and 2C. We corrected this mistake in the two new volcano plots of Figure 2C.

8) In the figure legend, Y axis is indicated as log10 of the p-values while in the Figure 2C, log (FDR). Clarify which one: Remake the Figure 2C so that two drug effects can be easily differentiated: For the vertical axis in Figure 2C, is it log(FDR) or log10FDR ? as typically, "log10FDR" is used. In Figure 2C, genes can be better illustrated with different colours for gene with logFC >0 and those <0, and different shape for the drugs, or separate volcano plots. For insignificant genes, use grey. Further, this figure is problematic to me; as it does not seem to reflect a typical volcano plot. If FDR used, then many of the non-significant p-values would be 1.0, and the Log of 1 is zero. So, many more of the values should be around zero. Some explanation needed.

We used the negative log_10_ (p values) in Figure 2C. We included in the revised version of the manuscript two separate volcano plots for each drug as recommended. We modified the respective legend accordingly. p values were used, and significance was set at 0.05. Significance in the volcano plots is indicated by a grey line that correspond to -log10 (0.05) = 1.3 in the new Figure 2C.

9) It is OK to leave as they are: For Figure 2(D) to (G), for each of these networks, how unusual is it to observe this? Given the bias, you are bound to have some networks, even by randomly selecting genes into IPA. Can the authors provide the p values or FDR to address this, and label them on the Figures directly?

As mentioned in response 2, the gene network identification in Figure 2D to 2G was based on each set of the differentially expressed genes observed between the treated and control groups. IPA used the Ingenuity Knowledge Base to screen for reported interactions involving these genes. The identified networks were scored based on the number of network eligible molecules they contain. The IPA network score for each gene network displayed as the negative log of the p-value of that specific network, gives the likelihood that the set of genes in this network could be explained by chance alone. Therefore, networks with a score ≥ 2 have at least 99% confidence that it is not generated by chance (27-30). Although a potential bias, the high network scores obtained for o-Vanillin (37, 24 and 13) and for RG-7112 (21) showed a strong degree of relevance to the Network Eligible Molecules in our dataset. The p-value, calculated using Fisher’s exact test, are typically quite low, it is visually easier to concentrate on the exponent and the p-score is defined as p-score = -log10(p-value). p scores (=network scores) are provided in Supplementary file 4. As recommended, we labeled the p scores of each network directly on the figures.

**References**

1) J. Campisi, Aging, cellular senescence, and cancer, Annual review of physiology 75 (2013) 685-705.

2) S. Parrinello, J.P. Coppe, A. Krtolica, J. Campisi, Stromal-epithelial interactions in aging and cancer: senescent fibroblasts alter epithelial cell differentiation, J Cell Sci 118(Pt 3) (2005) 485-96.

3) K. Tominaga, The emerging role of senescent cells in tissue homeostasis and pathophysiology, Pathobiology of Aging & Age Related Diseases 5 (2015).

4) J.C. Acosta, A. Banito, T. Wuestefeld, A. Georgilis, P. Janich, J.P. Morton, D. Athineos, T.W. Kang, F. Lasitschka, M. Andrulis, G. Pascual, K.J. Morris, S. Khan, H. Jin, G. Dharmalingam, A.P. Snijders, T. Carroll, D. Capper, C. Pritchard, G.J. Inman, T. Longerich, O.J. Sansom, S.A. Benitah, L. Zender, J. Gil, A complex secretory program orchestrated by the inflammasome controls paracrine senescence, Nature cell biology 15(8) (2013) 978-90.

5) H. Cherif, D.G. Bisson, P. Jarzem, M. Weber, J.A. Ouellet, L. Haglund, Curcumin and o-Vanillin Exhibit Evidence of Senolytic Activity in Human IVD Cells in vitro, J Clin Med 8(4) (2019).

6) Xu M, Pirtskhalava T, Farr JN, Weigand BM, Palmer AK, Weivoda MM, Inman CL, Ogrodnik MB, Hachfeld CM, Fraser DG et al. (2018) Senolytics improve physical function and increase lifespan in old age. Nature Med 24, 1246–1256.

7) Yousefzadeh MJ, Zhu Y, McGowan SJ, Angelini L, Fuhrmann‐Stroissnigg H, Xu M, Ling YY, Melos KI, Pirtskhalava T, Inman CL et al. (2018) Fisetin is a senotherapeutic that extends health and lifespan. EBioMedicine 36, 18–28.

8) Li, W., Qin, L., Feng, R., Hu, G., Sun, H., He, Y., & Zhang, R. (2019, Jul). Emerging senolytic agents derived from natural products. Mech Ageing Dev, 181, 1-6. A

9) O.H. Jeon, C. Kim, R.M. Laberge, M. Demaria, S. Rathod, A.P. Vasserot, J.W. Chung, D.H. Kim, Y. Poon, N. David, D.J. Baker, J.M. van Deursen, J. Campisi, J.H. Elisseeff, Local clearance of senescent cells attenuates the development of post-traumatic osteoarthritis and creates a pro-regenerative environment, Nature medicine 23(6) (2017) 775-781.

10) Tovar, C., Graves, B., Packman, K., Filipovic, Z., Higgins, B., Xia, M., Tardell, C., Garrido, R., Lee, E., Kolinsky, K., To, K. H., Linn, M., Podlaski, F., Wovkulich, P., Vu, B., & Vassilev, L. T. (2013, Apr 15). MDM2 smallmolecule antagonist RG7112 activates p53 signaling and regresses human tumors in preclinical cancer models. Cancer Res, 73(8), 2587-2597. https://doi.org/10.1158/0008-5472.CAN-12-2807

11) B.G. Childs, M. Gluscevic, D.J. Baker, R.M. Laberge, D. Marquess, J. Dananberg, J.M. van Deursen, Senescent cells: an emerging target for diseases of ageing, Nature reviews. Drug discovery 16(10) (2017) 718-735

12) Weber, L. (2010, Feb). Patented inhibitors of p53-Mdm2 interaction (2006 – 2008). Expert Opin Ther Pat, 20(2), 179-191. https://doi.org/10.1517/13543770903514129

13) A. Calcinotto, J. Kohli, E. Zagato, L. Pellegrini, M. Demaria, A. Alimonti, Cellular Senescence: Aging, Cancer, and Injury, Physiol Rev 99(2) (2019) 1047-1078.

14) B. Vu, P. Wovkulich, G. Pizzolato, A. Lovey, Q. Ding, N. Jiang, J.J. Liu, C. Zhao, K. Glenn, Y. Wen, C. Tovar, K. Packman, L. Vassilev, B. Graves, Discovery of RG7112: A Small-Molecule MDM2 Inhibitor in Clinical Development, ACS Med Chem Lett 4(5) (2013) 466-9.

15) M. Andreeff, K.R. Kelly, K. Yee, S. Assouline, R. Strair, L. Popplewell, D. Bowen, G. Martinelli, M.W. Drummond, P. Vyas, M. Kirschbaum, S.P. Iyer, V. Ruvolo, G.M. Gonzalez, X. Huang, G. Chen, B. Graves, S. Blotner, P. Bridge, L. Jukofsky, S. Middleton, M. Reckner, R. Rueger, J. Zhi, G. Nichols, K. Kojima, Results of the Phase I Trial of RG7112, a Small-Molecule MDM2 Antagonist in Leukemia, Clin Cancer Res 22(4) (2016) 868-76.

16) Feng, C., et al., Disc cell senescence in intervertebral disc degeneration: Causes and molecular pathways. Cell Cycle, 2016. 15(13): p. 1674-84

17) Abdulhameed Al-Ghabkari and Aru Narendran. Cancer Biotherapy and Radio pharmaceuticals.May 2019.252-257.http://doi.org/10.1089/cbr.2018.2732

18) Zhu Y, Tchkonia T, Pirtskhalava T, et al. The Achilles' heel of senescent cells: from transcriptome to senolytic drugs. Aging Cell. 2015; 14:644–58.

19) Zhu Y, Tchkonia T, Fuhrmann-Stroissnigg H, et al. Identification of a novel senolytic agent, navitoclax, targeting the BCl^-^2 family of anti-apoptotic factors. Aging Cell. 2016; 15:428–35.

20) Chang J, Wang Y, Shao L, Laberge RM, Demaria M, Campisi J, Janakiraman K, Sharpless NE, Ding S, Feng W et al. (2016) Clearance of senescent cells by ABT263 rejuvenates aged hematopoietic stem cells in mice. Nature Med 22, 78–83.

21) Jamie N. Justice, Anoop M. Nambiar, Tamar Tchkonia, Nathan K. LeBrasseur, Rodolfo Pascual, Shahrukh K. Hashmi, Larissa Prata, Michal M. Masternak, Stephen B. Kritchevsky, Nicolas Musi, James L. Kirkland. Senolytics in idiopathic pulmonary fibrosis: Results from a first-in-human, open-label, pilot study. EBioMedicine, 2019; DOI: 10.1016/j.ebiom.2018.12.052.

22) Laberge, 2018. Unit Dose of an MDM2 Inhibitor that Provides Long-Lasting Relief from Idiopathic Pulmonary Fibrosis and other Pulmonary Conditions by Selectively Removing Senescent Cells from the Lung Publication number: 20180303828 https://patents.justia.com/patent/20180303828.

23) Laberge, 2019. Therapy For Removing Senescent Cells And Treating Senescenceassociated Disease Using An Mdm2 Inhibitor Publication number: 20190343832

https://patents.justia.com/patent/20190343832.

24) Kirkland JL, Tchkonia T. Cellular Senescence: A Translational Perspective. EBioMedicine. 2017;21:21-28. doi:10.1016/j.ebiom.2017.04.013

25) Savli H, Szendröi A, Romics I, Nagy B. Gene network and canonical pathway analysis in prostate cancer: a microarray study. Exp Mol Med 2008; 40:176–185.

26) Li Y, Carrillo JA, Ding Y, He Y, Zhao C, Zan L, Song J. Ruminal transcriptomic analysis of grass-fed and grain-fed angus beef cattle. PLoS One 2015; 10:e0116437.

27) Yick, C. Y. (2013). The airway smooth muscle in asthma: More than meets the eye.

28) Baranzini SE, Galwey NW, Wang J, Khankhanian P, Lindberg R, Pelletier D, et al. Pathway and network-based analysis of genome-wide association studies in multiple sclerosis. Hum Mol Genet 2009; 18:2078–2090.

29) Yan-Fang T, Dong W, Li P, Wen-Li Z, Jun L, Na W, et al. Analyzing the gene expression profile of pediatric acute myeloid leukemia with real-time PCR arrays. Cancer Cell Int 2012; 12:40

30) Naito Y, Kuroda M, Mizushima K, Takagi T, Handa O, Kokura S, et al. Transcriptome analysis for cytoprotective actions of rebamipide against indomethacin-induced gastric mucosal injury in rats. J Clin Biochem Nutr 2007; 41:202–210.

31) Reyes-Gibby CC, Yuan C, Wang J, Yeung SC, Shete S. Gene network analysis shows immune-signaling and ERK1/2 as novel genetic markers for multiple addiction phenotypes: alcohol, smoking and opioid addiction. BMC Syst Biol 2015; 9:25.

32) Jia P, Kao CF, Kuo PH, Zhao Z. A comprehensive network and pathway analysis of candidate genes in major depressive disorder. BMC Syst Biol 2011; 5 (Suppl 3):S12.

33) https://www.rdocumentation.org/packages/stats/versions/3.6.2/topics/fisher.test